# An Integrated Co-Design Optimization Toolchain Applied to a Conjugate Cam-Follower Drivetrain System

Rocco Adduci [1,2,*], Jeroen Willems [3], Edward Kikken [3], Joris Gillis [2,4], Jan Croes [1,2] and Wim Desmet [1,2]

1   LMSD Research Group, Mechanical Engineering Department, KU Leuven University, 3000 Leuven, Belgium
2   DMMS Core Labs, Flanders Make, 3001 Heverlee, Belgium
3   Flanders Make, 3920 Lommel, Belgium; jeroen.willems@flandersmake.be (J.W.);
    edward.kikken@flandersmake.be (E.K.)
4   MECO Research Team, Department Mechanical Engineering, KU Leuven, 3000 Leuven, Belgium
*   Correspondence: rocco.adduci@kuleuven.be

**Abstract:** Due to ever increasing performance requirements, model-based optimization and control strategies are increasingly being adopted by machine builders and automotive companies. However, this demands an increase in modelling effort and a growing knowledge of optimization techniques, as a sufficient level of detail is required in order to evaluate certain performance characteristics. Modelling tools such as MATLAB Simscape have been created to reduce this modelling effort, allowing for greater model complexity and fidelity. Unfortunately, this tool cannot be used with high-performance gradient-based optimization algorithms due to obfuscation of the underlying model equations. In this work, an optimization toolchain is presented that efficiently interfaces with MATLAB Simscape to reduce user effort and the necessary skill and computation time required for the optimization of high-fidelity drivetrain models. The toolchain is illustrated on an industrially relevant conjugate cam-follower system, which is modelled in the Simscape environment and validated with respect to a higher-fidelity modeling technique, namely, the finite element method (FEM).

**Keywords:** optimal control; concurrent design; conjugate cam-follower; high-fidelity model; lumped parameter model; mechanical transmission; Simscape

## 1. Introduction

Nowadays, increasing requests for smart and high performance products are continuously challenging machine builders and vehicle drivetrain manufacturing companies to achieve higher throughputs, increased accuracy, reduced energy consumption, and improved comfort. At the same time, original equipment manufacturers (OEMs) are being challenged to reduce lead times, increase product customization, comply with strict environmental regulations, and deal with the competitive context of the globalised marketplace [1]. In this context, conventional prototype-based techniques are becoming ever more substituted by digitalized processes, as the former techniques are often time- and budget-consuming and rely strongly on the designer's ideas and knowledge.

Physics-based simulation models are increasingly being adopted to tackle these challenges, as they can provide versatile and modular processes which do not require deep knowledge and skills on the part of the end-user after being deployed in form of automated tools for designing hardware and/or tuning the controls of mechatronic systems [2–4], and can more easily be integrated at the different stages of the supply chain.

Nonetheless, such design procedures are often performed sequentially or iteratively [5], where the hardware is designed first and the control software afterwards [6]. Consequently, this yields sub-optimal designs [7] and update cycles with slow incremental improvements with respect to the final products. Co-design optimization approaches are commonly adopted to cope with these problems, where both parameters of hardware and controllers are optimized simultaneously [8,9]. Examples include co-design optimization

of hard disks [10], DC motors [11], active suspensions [12], four-bar linkages [13], etc. Furthermore, a distinction is made between nested (layered) and direct (simultaneous) approaches to solve the co-design problem [14]. In the first case, an outer loop optimizes the model/design parameters and an inner loop optimizes the controls. In the second case, both the model/design parameters and controls are solved in the same problem. In both cases, a system-level optimum can be achieved; however, in the second approach, although a more complex problem has to be solved, it is often possible to find the optimal solution more quickly. Despite their capabilities, co-design optimization techniques require: (1) accurate and computationally-efficient multi-physical models, and (2) expert knowledge of optimization techniques, which limits their industrial uptake.

In this work, a model-based optimization toolchain is presented which aims to overcome the above-described limitations, which we call the DriveTrain Co-Design Toolchain. It consists of the following main elements/steps:

(i) MATLAB Simscape [15,16] is used to create a 1D drivetrain model, permitting accurate and fast evaluation of the represented physics.

(ii) A pre-processor extracts the parametric Differential-Algebraic Equations (DAE) in symbolic form from Simscape. These white-box model equations can then be directly used in the gradient-based optimization problems listed next. They enable efficient (high-order) derivative evaluations through algorithmic differentiation [17]. This allows the optimization problems to run efficiently, eliminating the need for the use of (approximate) finite difference-based methods [18] or surrogate modelling [19] to derive the gradients [20].

(iii) Using the model, an Optimal Control Problem (OCP) is solved which optimizes the dynamic response and controls of the considered system according to a cost function and a set of constraints. A direct transcription of the OCP into a large-but-sparse Non-Linear Program (NLP) is performed using CasADi [17] and solved with IPOPT [21]. This approach scales well with horizon size and model order [22], and yields fast convergence.

(iv) A co-design optimization is performed in a single optimization problem (i.e., a direct approach); design parameters are added as additional degrees of freedom in the NLP that encodes the OCP. Although a more complicated problem has to be solved, it can yield the optimal results significantly faster compared to a nested approach ([23,24]), as the optimization problem has to be solved only once.

The complete toolchain is demonstrated on an industrially relevant conjugate cam-follower drivetrain. Cam-follower mechanisms are widely used in industry in many types of applications [25–28] because of their peculiar properties of transforming a given input motion into a desired one in the output. Many different types of cam designs have been studied in the literature depending on the application and requirements [29]. In [30], the conjugate nature of the cam-follower transfer unit were underlined as being of key importance in the system dynamics, reducing the required input energy and ensuring smooth transmission of motion between the driver and the driven mechanisms. Cam-follower mechanisms are most often modelled considering only the kinematics, with the dynamics introduced by the contact mechanism and body flexibilities being neglected; therefore, the model acts as an ideal transformer [27]. In practice, due to the flexibilities of the components and geometrical plays of the assembly chains, interacting bodies can lose contact under static and/or dynamic conditions, which might seriously jeopardise the performance and health of the system. In this contribution, to concurrently optimize the design and control of the mechanism, both states and design parameters are coupled in the modelling environment through an efficient motion parameterization. In this way, we limit the amount of design variables, analytically solve the inverse kinematics, and introduce an analytical nonlinear contact model. To achieve the industrially required model fidelity, i.e., accurate prediction of the contact forces, a customized Simscape Lumped Parameter Model (LPM) of a cam-follower system is developed and validated using a higher-fidelity nonlinear Finite Element Model (FEM).

The user-defined cam-follower model is then assembled into a complete Simscape drivetrain architecture, which is included and optimized in the optimization toolchain with little effort. In this context, the cam geometry and driving control torque are concurrently optimized. As a result, the system performance improvements are scored on a Pareto front, trading off relevant industrial concerns such as reducing the energy consumption with reducing the loads on the system (lifetime) by optimally choosing the design parameters and drivetrain control signals.

Our contribution is structured as follows: Section 2 describes the LPM approach we considered to mathematically represent the conjugate cam-follower system in the Simscape environment; in Section 3, the co-design toolchain structure is presented, moving from extraction of the symbolic equation to mathematical formulation of the co-design problem; in Section 4, the cam-follower LPM is first benchmarked with respect to its nonlinear FEM representation, then is brought into the toolchain, where multi-objective OCP and design problems are concurrently solved and the relevant results are illustrated; finally, Section 5 concludes this scientific contribution.

## 2. Cam-Follower Drivetrain Model

Cam-follower systems are widely used in the mechanical community, and often play a key role in the drivetrains of industrial machines. They can be understood as transformation mechanisms that convert the rotational motion of a driver component (cam) into a desired oscillating motion of another body (follower) by direct contact. Many fields of application can be found in the literature [25–28], from heavy to lightweight machines:

- Industrial and commercial machinery for goods and services, for example, shoe making, steel, and weaving mill equipment, as well as paper printing presses.
- Agricultural machinery and robotics for pick-and-place or cyclic operations.
- Microelectromechanical systems (MEMS) for accurate micromachinery in miniature control systems.
- Automotive performance and optimization, such as in high-speed automotive valve operating systems.

The widespread usage of such systems demands the constant evolution of design and system performance, forcing the analyst and design community to exploit physics-inspired simulation models in alternative to conventional prototype-based techniques. Additionally, these models can be used to efficiently predict dynamic system performance while being included in optimisation loops to simultaneously score the best design candidate which meets the desired targets.

In this work, an ad hoc 1D dual cam-follower model has been developed in MATLAB Simscape, permitting automatic code extraction as well as inclusion of the relevant physics in the system-level model. The Simscape implementation allows multi-physical component models to be directly integrated through block diagrams in the system-level model. The resulting modelling approach and interface permit us to concurrently employ an optimal control strategy minimizing the objective performances as well as to optimize the design parameters.

The system level architecture of the cam-follower shown in Figure 1 consists of the following main elements:

1. The input torque element;
2. Damping elements representing the input–output dissipating energy;
3. The inertia element of the conjugate cams and follower bodies;
4. The conjugate cam-follower (interaction) element.

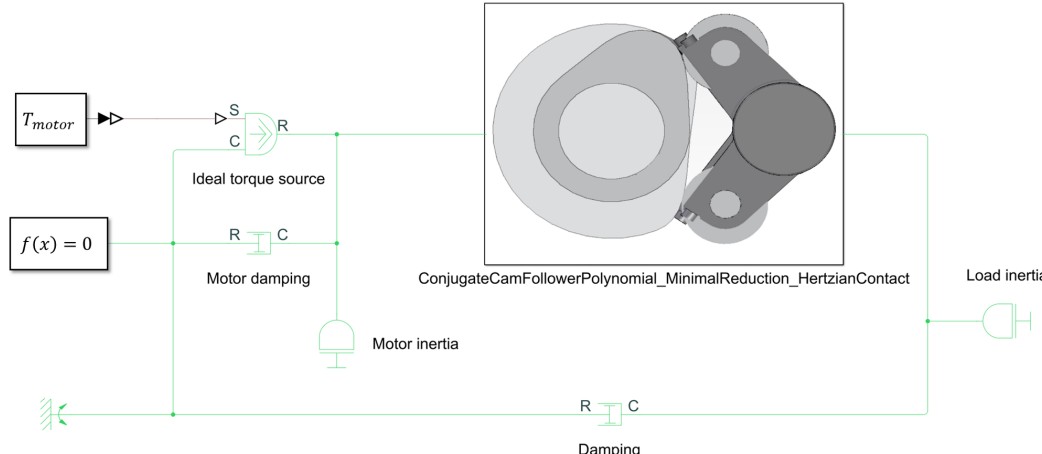

**Figure 1.** Simscape model implementing the conjugate cam-follower mechanism.

Each of the above elements are governed by a set of ordinary and/or algebraic differential equations automatically assembled via MATLAB Simscape based on the physical connections defined by block diagram.

The goal of this section is to provide the most relevant modelling elements to build an accurate system level model that accounts for the design parameters variations. For this reason, we focus on the cam-follower inertia parameterization and the differential equations of the conjugate cam-follower contact element, while referring to the Simscape documentation for more details of the remaining components.

### 2.1. The Cam and Follower Inertia Parameterization

Both conjugate cam and follower elements consist of a master ($M$) and slave ($S$) component connected to a rotational axis, where the inertial contributions can be expressed as follows:

$$J_{cam} = J_{cam,M} + J_{cam,S}; \tag{1a}$$

$$J_{follower} = J_{follower,M} + J_{follower,S}. \tag{1b}$$

Due to the bulky nature of both bodies, the influence of the geometrical parameters on the system inertia can be globally captured considering simplified/equivalent geometries. The cam shapes, for instance, can be reduced to circular disks of thickness $B_{cam}$ and radius $R_{eq}$, while the follower shapes can be considered as a combination of a beam of length $L_{follower}$ and a rectangular section of dimensions $R_{roller} \cdot B_{follower}$, with a point mass $M_{roller}$ at the end of the beam representing the roller body. For the single components $j = M, S$, the inertia contribution can be parameterized as

$$J_{cam,j} = \frac{\pi}{2}\rho_{cam,j}B_{cam,j}R_{eq,j}^4; \tag{2a}$$

$$
\begin{aligned}
J_{follower,j} = {} & \frac{1}{3}\rho_{follower,j}B_{follower,j}R_{roller,j}L_{follower,j}^3 + \\
& + \frac{1}{12}\rho_{follower,j}L_{follower,j}R_{roller,j}B_{follower,j}^3 + \\
& + M_{roller,j}L_{follower,j}^2
\end{aligned}
\tag{2b}
$$

where $\rho_{cam,j}$ and $\rho_{follower,j}$ are the cam and follower material density of the $j^{th}$ component. After the inertial contributions are parameterized with respect to the design parameters, they can be included in the rotational dynamic model depicted in Figure 1.

### 2.2. The Parameterized Cam-Follower Contact Element

Cam-follower mechanisms are most often modelled considering only the forward kinematics, where no contact loss is assumed and the kinematics are derived given a constant cam velocity [27]. The outputs of such analysis are the accelerations and relative positions of the centres of mass of both cam and follower, which in turn are fed into the dynamic force equilibrium equations. On the contrary, through the solution of the inverse kinematics, the motion is driven by the applied forces. If the assumption of no contact loss is considered, the model acts as an ideal transformer where the cam rotation is transformed to a oscillating motion at the follower side. However, in practice, due to the flexibilities of the components and geometrical plays of the assembly chains, interacting bodies can lose contact under static and/or dynamic conditions, which might seriously jeopardise the performance and health of the system. To this end, optimal design and control of the mechanism are required. In order to capture these effects in the modelling environment, the equations of motion governing the system dynamics must include both the state and design parameter dependencies.

In the following section, a motion parameterization is first introduced to limit the amount of design variables involved in the design optimization process while covering a wide range of solutions; next, an analytical nonlinear contact model is introduced; finally, both models are assembled and the dynamic equations of the cam-follower system are expressed.

#### 2.2.1. Piece-Wise Polynomial Follower Law

Starting from the kinematic relationship between the cam and the follower, the geometrical cam-follower design is defined through parameterization of the follower motion profile $\gamma$ and its partial derivatives with respect to the cam orientation $\theta$:

$$\gamma = f(\theta); \tag{3a}$$

$$\frac{d\gamma}{d\theta} = f_d(\theta); \tag{3b}$$

$$\frac{d^2\gamma}{d\theta^2} = f_{dd}(\theta). \tag{3c}$$

There are many ways to mathematically express the motion profile of Equation (3a)–(3c).

For instance, the follower motion can be described by a variety of functions depending on the amount of design parameters involved and the function complexity: *cycloid, modified harmonic, trapezoidal, modified trapezoidal, polynomial, spline, Bézier, harmonic*, etc. [29,31–35]. In practice, the profile shown in Figure 2 is one of the most commonly used in industry, namely, a dwell ($s_1$)–drop ($s_2$)–dwell ($s_3$)–rise ($s_4$) motion. It is schematized in different sectors $s$ and is mathematically represented by a *piece-wise polynomial* curve:

$$f(\theta) = \sum_s \left( p_{0,s} + p_{1,s} \frac{\Delta\theta_s}{\Delta\bar{\theta}_s} + p_{2,s} \frac{\Delta\theta_s}{\Delta\bar{\theta}_s}^2 + \dots + p_{n,s} \frac{\Delta\theta_s}{\Delta\bar{\theta}_s}^n \right); \tag{4a}$$

$$f_d(\theta) = \sum_s \left( \frac{p_{1,s}}{\Delta\bar{\theta}_s} + \frac{2p_{2,s}}{\Delta\bar{\theta}_s^2}\theta + \dots + \frac{np_{n,s}}{\Delta\bar{\theta}_s^n}\theta^{n-1} \right); \tag{4b}$$

$$f_{dd}(\theta) = \sum_s \left( \frac{2p_{2,s}}{\Delta\bar{\theta}_s^2} + \dots + \frac{n(n-1)p_{n,s}}{\Delta\bar{\theta}_s^n}\theta^{n-2} \right). \tag{4c}$$

where $n$ represents the polynomial order and

$$\Delta\theta_s = (\theta - \bar{\theta}_s); \tag{5a}$$

$$\Delta\bar{\theta}_s = (\bar{\theta}_s - \bar{\theta}_{s-1}). \tag{5b}$$

$\bar{\theta}_s$ and $\bar{\theta}_{s-1}$ are the cam angle limits of each sector $s$.

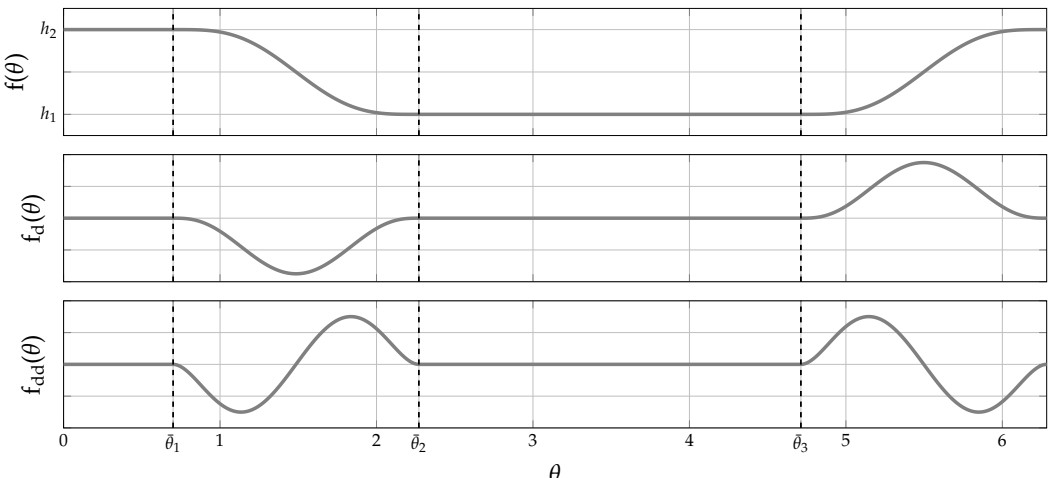

**Figure 2.** Follower kinematics motion: dwell–drop–dwell–rise.

In this work, the polynomial order is set to $n = 7$ for a smooth representation of the follower law and its derivatives with respect to the cam angle in each sector, as well as to analytically derive the polynomial parameters $p_{n,s}$ while assuming a minimal set of boundary conditions, as summarized in Table 1.

As result, the designed follower law curve represents a cam-follower geometry, as shown in Figure 3, and the choice of the polynomial description allows us to cover a wide and industrially relevant design space while minimizing the amount of design parameters.

**Table 1.** Seventh-order polynomial parameter values.

| | Polynomial Parameters | | | | | | | |
|---|---|---|---|---|---|---|---|---|
| **Sectors** | $p_0$ | $p_1$ | $p_2$ | $p_3$ | $p_4$ | $p_5$ | $p_6$ | $p_7$ |
| 1 | $h_1 + h_2$ | 0 | 0 | 0 | 0 | 0 | 0 | 0 |
| 2 | $h_1 + h_2$ | 0 | 0 | 0 | $-35\,h_2$ | $84\,h_2$ | $-70\,h_2$ | $20\,h_2$ |
| 3 | $h_1$ | 0 | 0 | 0 | 0 | 0 | 0 | 0 |
| 4 | $h_1$ | 0 | 0 | 0 | $35\,h_2$ | $-84\,h_2$ | $70\,h_2$ | $-20\,h_2$ |

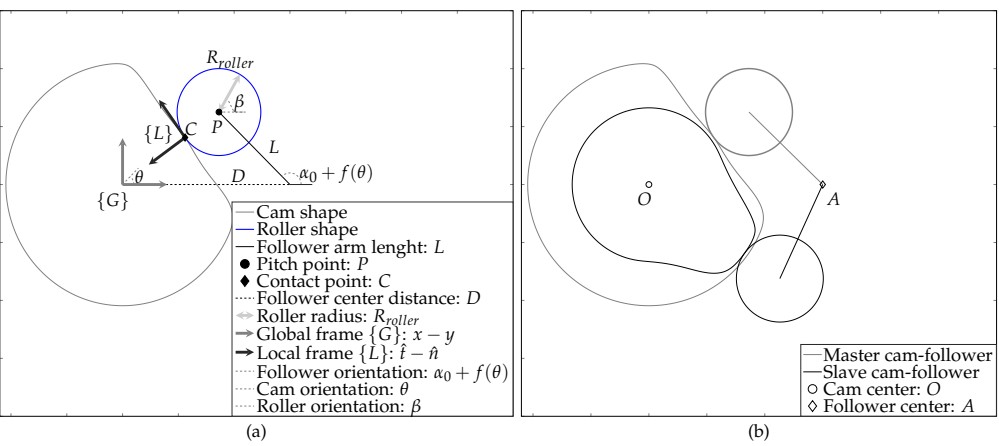

**Figure 3.** (**a**) Cam-follower nomenclature and (**b**) conjugate cam-follower representation.

### 2.2.2. Cam-Follower: Non-Linear Contact Force Model

In order to account for the dynamic effects introduced by the cam-follower interactions, a nonlinear force model is introduced acting in a local frame $\{L\}$ with unit vectors $\hat{t} - \hat{n}$, as shown in Figure 3a. Here, the roller contact point $C$ is allowed to slide over the local

normal axis frame, enabling the contacting bodies to overlap each other and resulting in a local penetration. Because the latter is much smaller than the bodies' dimensions, the displacement in the tangential direction can be neglected as long as the center points of the cam and follower are considered fixed. Thanks to the definition of the local contact forces, different contact models with variable complexity/fidelity can be plugged and played at the system level without modifying the overall model architecture. In particular, the normal contact forces are obtained through a nonlinear stiffness–displacement relationship expressed as a function of the penetration and its time derivative. The normal force $F_n$ acting on each of the interacting bodies is defined as the sum of the elastic $F_{n,e}$ and viscous $F_{n,v}$ contributions:

$$F_n = F_{n,e} + F_{n,v}. \tag{6}$$

$F_{n,e}$ is computed assuming that the main deformation field of the interacting bodies occurs in the contact area and that the contact patch between the cam and roller is a line. This allows us to model the contact phenomena by applying the Hertzian theory [36] for cylindrical bodies on a contact segment of length $B$. The nonlinear contact deformation $\delta$ that arises from the load $F_{n,e}$ can be described with the closed-form formula derived by Weber and Banaschek in [37].

$$\delta = \frac{F_{n,e}}{\pi B} \left( \frac{1 - v_{cam}^2}{E_{cam}} + \frac{1 - v_{roller}^2}{E_{roller}} \right) \left[ ln \left( \frac{4 h_{cam} h_{roller}}{a^2} \right) - \frac{1}{3} \left( \frac{v_{cam}}{1 - v_{cam}} + \frac{v_{roller}}{1 - v_{roller}} \right) \right]. \tag{7}$$

Equation (7) assumes that the cam and roller can be well represented as cylinders with radii of curvature $R_{cam}$ and $R_{roller}$, respectively, in the near proximity of the contact area. In this regard, while $R_{roller}$ is constant, $R_{cam}$ is a function of the cam orientation ($\theta$), as discussed in Appendix A. The parameters $h_{cam}$ and $h_{roller}$ are chosen to be equal to $R_{cam}$ and $R_{roller}$, respectively. Hertz derived the analytical formulas that allow the half contact width $a$ and the maximum contact pressure $\kappa$ to be computed as functions of a contact load $F_{n,e}$ [38]:

$$a = \left[ \frac{4 F_{n,e}}{\pi B} R^* \left( \frac{1 - v_{cam}^2}{E_{cam}} + \frac{1 - v_{roller}^2}{E_{roller}} \right) \right]^{1/2}; \tag{8a}$$

$$\kappa = \frac{2 F_n}{\pi a B}. \tag{8b}$$

Here, the parameter $B$ is defined as the overlapping thickness of the interacting bodies, that is, the cam and roller. The material characteristics of the cam and roller are taken into account through the Young's modulus $E$ and Poisson ratio $v$. The equivalent radius of curvature $R^*$ accounts for the relative curvature of the two cylinders at the contact point:

$$\frac{1}{R^*} = \frac{1}{R_{cam}} + \frac{1}{R_{roller}}. \tag{9}$$

Subsequently, the nonlinear viscous contribution $F_{n,v}$ can be considered as only a repulsive damping force (the bodies are pushed away from each other) when the bodies are in contact; it reads as

$$F_{n,v} = C_v |\dot{\delta}|. \tag{10}$$

Finally the tangent forces $F_t$, represented as a frictional contribution occurring along the tangent direction $\hat{t}$, are a combination of the static and kinetic Coulomb friction models:

$$F_t = sign(\dot{\theta}) \begin{cases} \mu_s F_n, & \text{if } v_{slip} = 0 \\ \mu_d F_n, & \text{otherwise} \end{cases}. \tag{11}$$

where $\mu_s$ and $\mu_d$ are the static and dynamic friction coefficients, respectively. The expression $sign(\dot{\theta})$ has been included due to the dependency of the tangent force with respect to the angular cam velocity, as conventionally the normal force is assumed to be either positive or zero.

2.2.3. The Conjugate Cam-Follower Dynamic Equations

In light of the kinematic and contact force description provided above, the differential equations governing the conjugate cam-follower element are derived from contact theory [36] based on the velocity equilibrium $\vec{v}_c$ at the contact point and the torque balance equations:

$$\overrightarrow{v}_c \cdot \hat{n} = 0; \tag{12}$$

$$T_{cam} = \begin{bmatrix} OC'_t \\ OC'_n \end{bmatrix} \times \begin{bmatrix} F_t(v_{slip}, F_n, p) \\ F_n(\delta, \dot{\delta}, p) \end{bmatrix}; \tag{13a}$$

$$T_{follower} = \begin{bmatrix} AC'_t \\ AC'_n \end{bmatrix} \times \begin{bmatrix} F_t(v_{slip}, F_n, p) \\ F_n(\delta, \dot{\delta}, p) \end{bmatrix}. \tag{13b}$$

Here, the subscripts $t$ and $n$ stand for the projections on the $\hat{t} - \hat{n}$ local frame $\{L\}$, as indicated in Figure 3. Accordingly, $F_t$ and $F_n$ are the tangential and normal forces acting on each body, and are expressed as function design parameters $p$ representing the shapes of the cam and follower pairs, while $v_{slip}$ is the roller slip velocity and $\delta$ and $\dot{\delta}$ are the penetration and its time derivatives, respectively.

The local deformation $\delta$ is defined as the difference between the dynamic variable $AC'$ projected in direction $\hat{n}$ and the equivalent kinematic distance $AC$, computed by solving the inverse kinematics (reported in Appendix A). Thus, the resulting penetration $\delta$ can be written as

$$\delta = AC'_n - AC_n; \tag{14a}$$

$$\dot{\delta} = A\dot{C}'_n - A\dot{C}_n. \tag{14b}$$

Subsequently, the differential equation governing the roller dynamics is introduced as follows:

$$J_{roller}\ddot{\beta} + C_{bearing}\dot{\beta} + R_{roller}F_t = 0, \tag{15}$$

where $J_{roller}$ is the roller inertia, $C_{bearing}$ is the bearing damping, $R_{roller}$ is the roller radius, and $\beta$ is the roller angular orientation (state) expressed in the roller local frame. Finally, the roller slip velocity can be computed as

$$v_{slip} = R_{roller}\dot{\beta} - O\dot{C}_t. \tag{16}$$

## 3. DriveTrain Co-Design Toolchain

In this section, an overview of the developed DriveTrain Co-Design Toolchain is provided. It is designed to easily set up the required optimization problems starting from a Simscape model using only a handful of high-level commands. The low-level code is abstracted from the user, and is efficiently handled by a back-end that interfaces with CasADi's Opti-stack [39]. The toolchain consists of the following steps:

1.  Creation of a high-fidelity drivetrain model in MATLAB Simscape.
2.  Extraction of symbolic equations from the Simscape model using the developed tool `Simscape2CasADi`. More information about this tool is provided in Section 3.1.
3.  Formulation of the optimization problems. A user-friendly interface is provided to set up the optimization problems. Two types of optimization problems are considered:

    *   Optimal control; see Section 3.2. In this case, the controls of the model are optimized according to a user-defined cost function and set of constraints.

- Concurrent design (co-design); see Section 3.3. In this case, both the controls and specific model parameters are optimized according to a cost function and a number of constraints defined by the user.

More details about the practical steps required to set up the optimization problems are provided later in this section.

### 3.1. Simscape2CasADi: Extracting Symbolic Equations from Simscape

This part of the toolchain extracts equations from Simscape models using the tool `Simscape2CasADi` [40]. Parametric, time-dependent, time-delayed, and nonlinear models are all supported, even with if-tests, though not models involving a finite-state machine. The output is a set of differential algebraic equations (DAE) in the following form:

$$\dot{x}(t) = f_{ode}(x(t), z(t), u(t), p); \tag{17a}$$

$$0 = f_{alg}(x(t), z(t), u(t), p); \tag{17b}$$

$$y(t) = f_{out}(x(t), z(t), u(t), p), \tag{17c}$$

with $t \in \mathbb{R}$ being the time, $x \in \mathbb{R}^{n_x}$ the differential state variables, $z \in \mathbb{R}^{n_z}$ the algebraic variables, $u \in \mathbb{R}^{n_u}$ the inputs, $p \in \mathbb{R}^{n_p}$ the parameters, and $y \in \mathbb{R}^{n_y}$ the output vector.

In addition to symbolic descriptions of $f_{ode}$, $f_{alg}$ and $f_{out}$, the tool provides metadata (names) for the variables and parameters. The tool involves four steps:

1. C-code generation is performed on the Simulink model;
2. The Simscape part of the C-code is parsed;
3. A MATLAB class that implements $f_{ode}$, $f_{alg}$ and $f_{out}$ using CasADi [17] symbols is created;
4. The index of the DAE is optionally reduced with the help of MATLAB's Symbolic Toolbox.

The extracted model includes symbolic (white-box) equations, which enable efficient (high-order) derivative evaluation through algorithmic differentiation [17].

### 3.2. Optimal Control

In this section, we detail the formulation of an optimal control problem. First, we provide the formulation for a continous-time optimal control problem. We focus on the direct approach for solving such optimal control problems [41]. In this case, the control trajectory is parametrized by a finite number of unknowns, forming a non-linear program (NLP). The general notation of such an NLP is provided and the system dynamics (see Section 3.1) are discretized. Lastly, we provide the resulting discrete-time optimal control problem.

#### 3.2.1. Continuous-Time Optimal Control Problem

The generic continuous-time optimal control problem on the horizon $t \in [0, T]$ (with $T$ being the final time) is defined using the model described in Section 3.1 as follows:

$$\underset{X, Z, U, Y}{\text{minimize}} \quad \mathcal{J}(X, Z, U, Y, p) \tag{18a}$$

$$\text{subject to} \quad \dot{x}(t) = f_{ode}(t, x(t), z(t), u(t), p) \ \forall \, t \in [0, T]; \tag{18b}$$

$$0 = f_{alg}(t, x(t), z(t), u(t), p) \ \forall \, t \in [0, T]; \tag{18c}$$

$$y(t) = f_{out}(t, x(t), z(t), u(t), p) \ \forall \, t \in [0, T]; \tag{18d}$$

$$\underline{g} \leq g(X, Z, U, Y, p) \leq \bar{g}. \tag{18e}$$

In the above, $X$, $Z$, $U$, and $Y$ denote the entire trajectories over horizon $t \in [0, T]$. Equation (18a) can be used to implement various costs, such as the well-known Lagrange term (way-cost) and Mayer term (end-cost) [17]. Equation (18e) can be used to implement multiple types of constraints, such as initial condition constraints, input and path constraints, boundary constraints, etc.

### 3.2.2. Transcription of the Continuous OCP to a Discrete-Time OCP

In this section, the formulation of the considered discrete-time optimal control problem is detailed, with particular focus on a direct approach to solve optimal control problems. The basic idea of such a direct method is to parameterise the trajectories of the OCP by a finite number of unknowns, resulting in a nonlinear program of finite dimensions [41]:

$$\underset{\boldsymbol{w}_{opt}}{\text{minimize}} \qquad \mathcal{J}(\boldsymbol{w}_{opt}) \tag{19a}$$

$$\text{subject to} \qquad \underline{\boldsymbol{g}} \leq \boldsymbol{g}(w_{opt}) \leq \bar{\boldsymbol{g}}, \tag{19b}$$

where $\mathcal{J} \in \mathbb{R}$ denotes the cost function and $\boldsymbol{w}_{opt} \in \mathbb{R}^{n_w}$ denotes the (list of) continuous optimization variables. Furthermore, $\boldsymbol{g} \in \mathbb{R}^{n_g}$ denotes arbitrary constraint functions, with $\underline{\boldsymbol{g}} \in \mathbb{R}^{n_g}$ and $\bar{\boldsymbol{g}} \in \mathbb{R}^{n_g}$ the respective lower and upper bounds.

In order to cast the continuous-time optimal control problem from Equation (18a)–(18e) in the structure of Equation (19a) and (19b), we first parameterize the time interval $t \in [0, T]$ in $N$ number of steps (samples) with a sampling time (or integration horizon) $T_s \in \mathbb{R}$. Additionally, we discretize the model dynamics, Equation (17a)–(17c), for which the result is provided by

$$\boldsymbol{x}_{k+1} = \boldsymbol{f}_k(\boldsymbol{x}_k, \boldsymbol{z}_k, \boldsymbol{u}_k, \boldsymbol{p}); \tag{20a}$$

$$\boldsymbol{0} = \boldsymbol{f}_{alg}(\boldsymbol{x}_k, \boldsymbol{z}_k, \boldsymbol{u}_k, \boldsymbol{p}); \tag{20b}$$

$$\boldsymbol{y}_k = \boldsymbol{f}_{out}(\boldsymbol{x}_k, \boldsymbol{z}_k, \boldsymbol{u}_k, \boldsymbol{p}), \tag{20c}$$

where $k \in [1, N]$ denotes the current time sample (yielding $\boldsymbol{x}_k \in \mathbb{R}^{n_x}$, $\boldsymbol{z}_k \in \mathbb{R}^{n_z}$, $\boldsymbol{u}_k \in \mathbb{R}^{n_u}$, $\boldsymbol{y}_k \in \mathbb{R}^{n_y}$). Note that the model parameters $\boldsymbol{p} \in \mathbb{R}^{n_p}$ are assumed to be independent of $k$. We introduce function $\boldsymbol{f}_k$ in Equation (20a), which denotes the state propagation from $k$ to $k+1$. For the time-discretized transition function $\boldsymbol{f}_k$, we can employ several integration schemes to approximate $\boldsymbol{x}_{k+1}$, such as forward Euler:

$$\tilde{\boldsymbol{x}}_{k+1} \approx \boldsymbol{x}_k + T_s \boldsymbol{f}_{ode}(\boldsymbol{x}_k, \boldsymbol{z}_k, \boldsymbol{u}_k, \boldsymbol{p}) = \boldsymbol{f}_{k+1}, \tag{21}$$

or backward Euler,

$$\tilde{\boldsymbol{x}}_{k+1} \approx \boldsymbol{x}_k + T_s \boldsymbol{f}_{ode}(\boldsymbol{x}_{k+1}, \boldsymbol{z}_{k+1}, \boldsymbol{u}_k, \boldsymbol{p}) = \boldsymbol{f}_{k+1}, \tag{22}$$

or various other schemes, such as fourth-order Runge–Kutta, direct collocation, etc. We can collect the resulting continuous optimization variables for the defined time grid $k \in [1, N]$ in matrices $\boldsymbol{X} \in \mathbb{R}^{n_x \times N}$, $\boldsymbol{Z} \in \mathbb{R}^{n_z \times N}$, $\boldsymbol{U} \in \mathbb{R}^{n_u \times N}$, and $\boldsymbol{Y} \in \mathbb{R}^{n_y \times N}$, in the form of $\boldsymbol{X} = [\boldsymbol{x}_0, \boldsymbol{x}_1, \ldots, \boldsymbol{x}_{N-1}, \boldsymbol{x}_N]$, and similarly for $\boldsymbol{z}$, $\boldsymbol{u}$, and $\boldsymbol{y}$. The resulting discrete-time optimal control problem is then provided by

$$\underset{\boldsymbol{X}, \boldsymbol{Z}, \boldsymbol{U}, \boldsymbol{Y}}{\text{minimize}} \qquad \mathcal{J}(\boldsymbol{X}, \boldsymbol{Z}, \boldsymbol{U}, \boldsymbol{Y}, \boldsymbol{p}) \tag{23a}$$

$$\text{s.t.} \quad \boldsymbol{x}_{k+1} = \boldsymbol{f}_{k+1}(\boldsymbol{x}_k, \boldsymbol{z}_k, \boldsymbol{u}_k, \boldsymbol{p}) \qquad \forall\, k \in [1, N]; \tag{23b}$$

$$\boldsymbol{0} = \boldsymbol{f}_{alg}(\boldsymbol{x}_k, \boldsymbol{z}_k, \boldsymbol{u}_k, \boldsymbol{p}) \qquad \forall\, k \in [1, N]; \tag{23c}$$

$$\boldsymbol{y}_k = \boldsymbol{f}_{out}(\boldsymbol{x}_k, \boldsymbol{z}_k, \boldsymbol{u}_k, \boldsymbol{p}) \qquad \forall\, k \in [1, N]; \tag{23d}$$

$$\underline{\boldsymbol{g}} \leq \boldsymbol{g}(\boldsymbol{X}, \boldsymbol{Z}, \boldsymbol{U}, \boldsymbol{Y}, \boldsymbol{p}) \leq \bar{\boldsymbol{g}}. \tag{23e}$$

In particular, Equation (23a) implements the cost function while Equation (23b)–(23d) embed the model dynamics in the optimization problem. The state propagation of Equation (23b) is implemented using a multiple shooting approach, as it scales better with longer horizons $N$ [42]. Furthermore, Equation (23e) can be used to implement multiple types of constraints, such as initial condition constraints, input and path constraints,

boundary constraints, etc. Here, $\underline{\square}$ and $\bar{\square}$ denote lower and upper bounds, respectively. The resulting discrete-time optimization problem is a large-but-sparse nonlinear program (NLP) containing continuous optimization variables, and is solved using IPOPT [21].

**Remark 1.** *The derived control input is a (non-parameterized) feedforward input signal (e.g., motor torque or force). This input can be applied directly to the considered system; however, for experimental implementation, typically a feedback controller is considered as well, providing robustness for non-modelled components and disturbances. This feedback controller then tracks a reference signal (e.g., the desired motor position or speed), which can be a direct outcome of the optimization problem as well. Alternatively, the found feedforward torque control law can serve as a benchmark for some other well-tuned control laws, ranging from simple PID controllers to parameterized feedforwards [43], as well as for model-predictive control (MPC) techniques [44]. Alternatively, if already available, such parameterizations of the control structure can be directly embedded in the optimization problem and then solved through adding additional constraints.*

### 3.3. Concurrent Design with Optimal Control

As introduced in the previous section, instead of only optimizing the controls for a given design (with model parameters $p$ being fixed), the controls and specific model parameters can be optimized at the same time, which is called concurrent design (or co-design). In this case, the cost function is provided by

$$\underset{X,Z,U,Y,p}{\text{minimize}} \qquad \mathcal{J}(X,Z,U,Y,p) \qquad (24)$$

Note that this equation differs from Equation (23a) since several model parameters $p$ are no longer fixed and become optimization variables. For these variables, we can introduce additional constraints

$$\underline{p} \leq p \leq \bar{p}, \qquad (25)$$

or constraints in the form of Equation (23e). The given co-design problem is solved in a single optimization problem (i.e., a direct co-design is adopted), yielding fast convergence.

### 3.4. Implementation in the Toolchain

The developed toolchain is designed to easily set up the required optimization problems, requiring only high-level commands and relying on CasADi's Opti-stack [39]. The practical steps required to set up and use the toolchain are shown schematically in Figure 4, and are further detailed below.

1.  First, we initialize the Matlab class implementing the DriveTrain Co-Design toolchain.
2.  The second step is to supply a model to the toolchain. A manually derived model in the from of a DAE can be supplied. Alternatively, the proposed equation extraction from Simscape (using `Simscape2CasADi`) can be used. In the latter case, the user provides a Simulink model that includes a parameter file. Afterwards, the extracted model is returned as a Matlab structure that contains the extracted DAE model as a CasADi function (implementing Equation (17a)–(17c)) and a vector denoting the names of the extracted states, inputs, outputs, and parameters.
3.  Next, an OCP/co-design problem is initialized. The Matlab structure obtained in the previous step can then be directly loaded in by the toolchain to embed the model dynamics.
4.  If a co-design problem is considered, the user has to define which model parameters have to be optimized, along with their lower and upper bounds. If an OCP is considered, this option can be skipped.
5.  Next, the number of samples $N$ and the sampling time $T_s$ have to be defined, along with a transcription method of choice:

- Multiple shooting with an integrator of choice (e.g., forward Euler, backward Euler, etc.);
- Direct collocation with a degree of choice.

This automatically creates the discrete optimization variables ($X$, $Z$, $U$, $Y$, and optionally $p$), and automatically implements the constraint Equation (23b)–(23d) and optionally Equation (25).

6. Using the above optimization variables, the scalar cost function $\mathcal{J}$ can be defined in the form of Equation (23a) or Equation (24). Miscellaneous constraints can then be provided by providing the term to be constrained, including the lower and/or upper bounds (according to Equation (23e)).
7. Lastly, the optimization problems are solved (by default, using IPOPT [21]), and the results are returned in a Matlab structure, which can then be visualized.
   - Prior to solving, an initial guess for the optimization variables can be provided by the user in order to enhance the convergence of the solver.

With the optimization problems defined, we now apply them to an industrially relevant case, namely, a cam-follower drivetrain system.

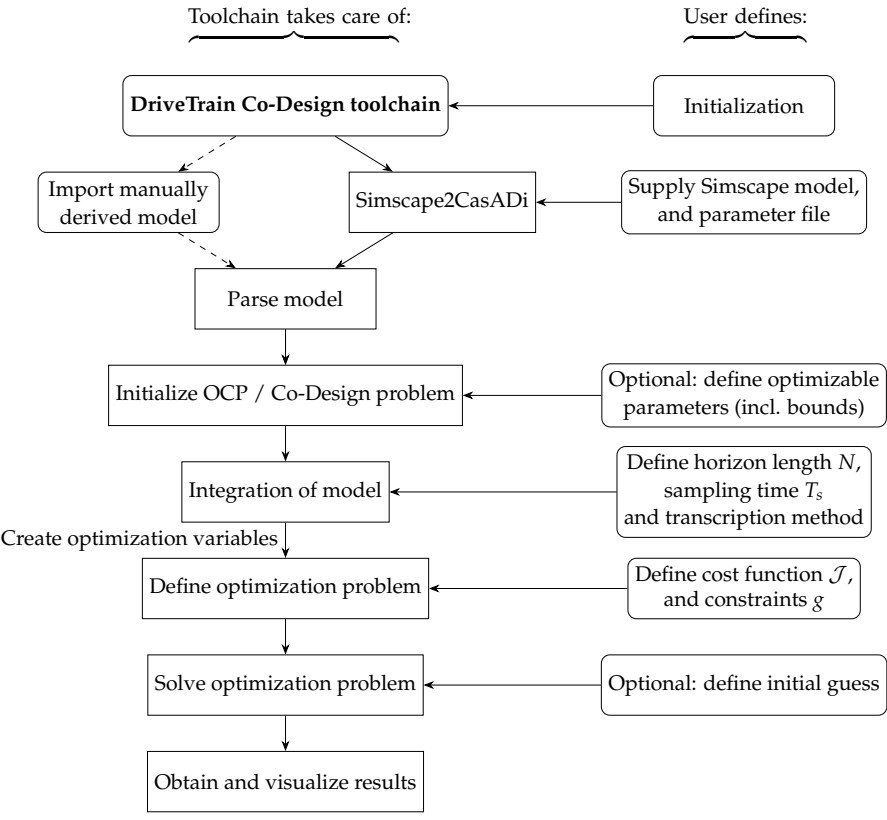

**Figure 4.** Flowchart describing the implementation of the DriveTrain Co-Design toolchain.

## 4. Model Validation, Toolchain Application, and Results

In this section, the model derived in Section 2 is first validated and subsequently applied to the proposed toolchain described in Section 3.

For these purposes, a reference conjugate cam-follower geometry is introduced. In Tables 2–4, the reference design and material parameters of an industrially relevant design are reported, and in Figure 3 the generated geometries are shown.

The resulting model is statically validated against a higher-fidelity nonlinear FE model in the next session to show the accuracy and limitations deriving from the adopted modelling assumptions. Finally, the system level cam-follower model is applied to both OCP and co-design problems.

**Table 2.** Reference model design parameters.

| Symbol | Master Value | Slave Value | Unit |
|--------|--------------|-------------|------|
| $R_{roller}$ | $37.5 \times 10^{-3}$ | $37.5 \times 10^{-3}$ | m |
| $D$ | $150 \times 10^{-3}$ | $150 \times 10^{-3}$ | m |
| $L$ | $89.2 \times 10^{-3}$ | $89.2 \times 10^{-3}$ | m |
| $B$ | $34.5 \times 10^{-3}$ | $34.5 \times 10^{-3}$ | m |
| $\alpha_o$ | 1.96 | −2.40 | rad |

**Table 3.** Reference follower law parameters.

| Symbol | Value | Unit |
|--------|-------|------|
| $h_1$ | 0 | rad |
| $h_2$ | 0.4 | rad |
| $\bar{\theta}_{1:4}$ | [0 1.12 4.04 5.16] | rad |

**Table 4.** Cam and follower mechanical properties.

| Symbol | Value | Unit |
|--------|-------|------|
| $\rho$ | 7829 | kg/m$^3$ |
| $E$ | $200 \times 10^3$ | MPa |
| $\nu$ | 0.3 | - |

### 4.1. Conjugate Cam-Follower Model Validation

In this subsection, the analytical contact model accuracy is compared to its nonlinear twin FE model. This analysis is carried out under static conditions, as the system-level dynamics are dominated by the local contact flexibility due to the bulky cam and follower bodies. First, the FE model representation of the cam-follower system is defined in the Abaqus/Standard environment and solved through a Newton-based iterative method. Second, a metric is defined to compare the model simulation results.

Here, the master and slave cam geometries are generated in the Abaqus environment by importing the cloud of points of the cam profiles created in MATLAB, then converted to spline curves in Abaqus. Finally, the 3D geometries are defined by extruding the cam profiles, while the the roller geometry is fully designed using the Abaqus interface, as shown in Figure 5. After the CAD files are generated, a linear hexamesh is assigned to the different bodies (see Figure 5) after a mesh convergence analysis.

To reduce computational effort and required memory allocation, the master and slave cam-follower subsystems are treated as independent simulations. Each internal surface of the cam and roller is constrained to the so-called Multi-Point Constraints ($MPC_{cam}$ and $MPC_{follower}$). In particular, the $MPC_{cam}$ is fixed to the world (six constraints), while $MPC_{follower}$ is constrained in five of the six degrees of freedom (dofs), allowing it to rotate around the follower axes, where several input torque values are sequentially applied. Finally, the contact between the cam and follower is detected between the interaction surfaces through surface-to-surface contact detection and by using the frictionless augmented Lagrange contact formulation.

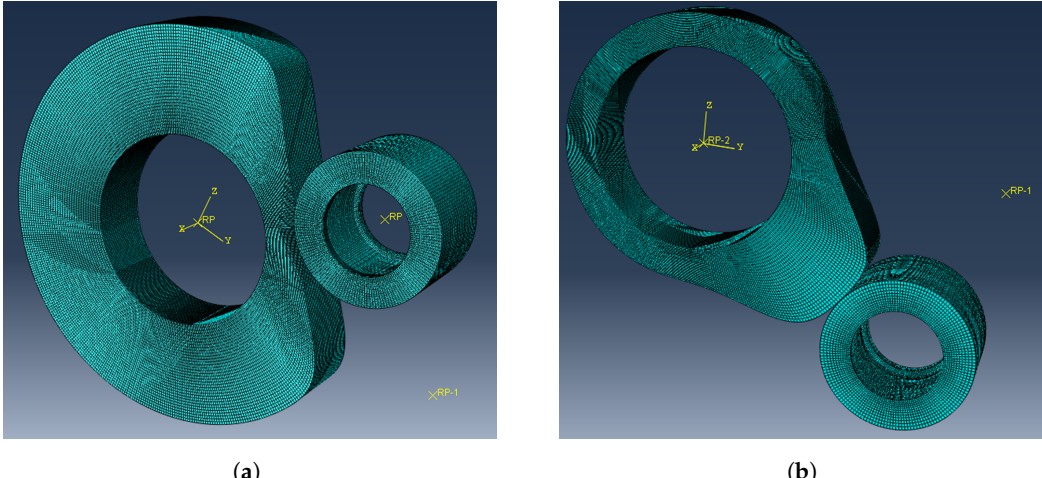

|  (a)  |  (b)  |

**Figure 5.** Abaqus models: (**a**) master cam and roller and (**b**) slave cam and roller.

Both LPM and FEM are evaluated in terms of the angular displacement $\psi'$ of the follower arm (output) resulting from a constant torque (input) applied on the follower axes for both of the master and slave subsystems shown in Figure 5. In order to appreciate the quality of the LPM with respect to the FEM, the static transmission error $STE$ is introduced as a performance metric; it is defined as the difference between the kinematic follower angle $\psi$ and follower angle computed at the static equilibrium $\psi'$ for each discrete cam configuration $\theta_i$:

$$STE(\theta_i) = \psi'(\theta_i) - \psi(\theta). \tag{26}$$

In Figure 6, the STE resulting from both the nonlinear LPM and the nonlinear FEM model are compared for seven different torque levels and several angular configurations. Despite the modeling simplifications, it is shown that the LPM is well able to predict the STE trends and evolution introduced by the nonlinear local compliance in the contact area compared to the higher-fidelity FEM solution. Figure 7 shows the absolute value of the relative STE error between the two modelling approaches, where it can be observed that an overall modeling error below 15% is obtained for the LPM with respect to the FEM, while a greater STE mismatch can be seen at the graph sides (at high pressure angle) due to the unmodeled body compliance. This is because the LPM only accounts for the local contact stiffness, which remains overall dominant. Moreover, the higher STE error observed at low torques is due to the relatively low values of the STE, which renders the relative error percentage more sensitive to small numbers. In general, it can be concluded that the LPM establishes a good trade-off between modeling accuracy and computational cost with respect to the FEM approach. These aspects are crucial to ensuring (i) the robustness of the entire toolchain, as the outcome of the co-design problem is highly dependent on the model fidelity/accuracy, and (ii) that the solutions of the optimization problems are reached in a time-efficient manner.

*4.2. Cam-Follower Model Equation Extraction*

The created Simscape model shown in Figure 1, including the conjugate cam-follower mechanism, is employed in the proposed co-design toolchain. Using the `Simscape2CasADi` tool, symbolic equations are extracted in the form of a set of differential algebraic equations of index 2 according to the procedure discussed in Section 3. The resulting model contains eight differential states, eleven algebraic states, one input, and five outputs. After the equation extraction, an integration check is performed to confirm that the dynamics of the extracted model match those of the original Simscape model [40].

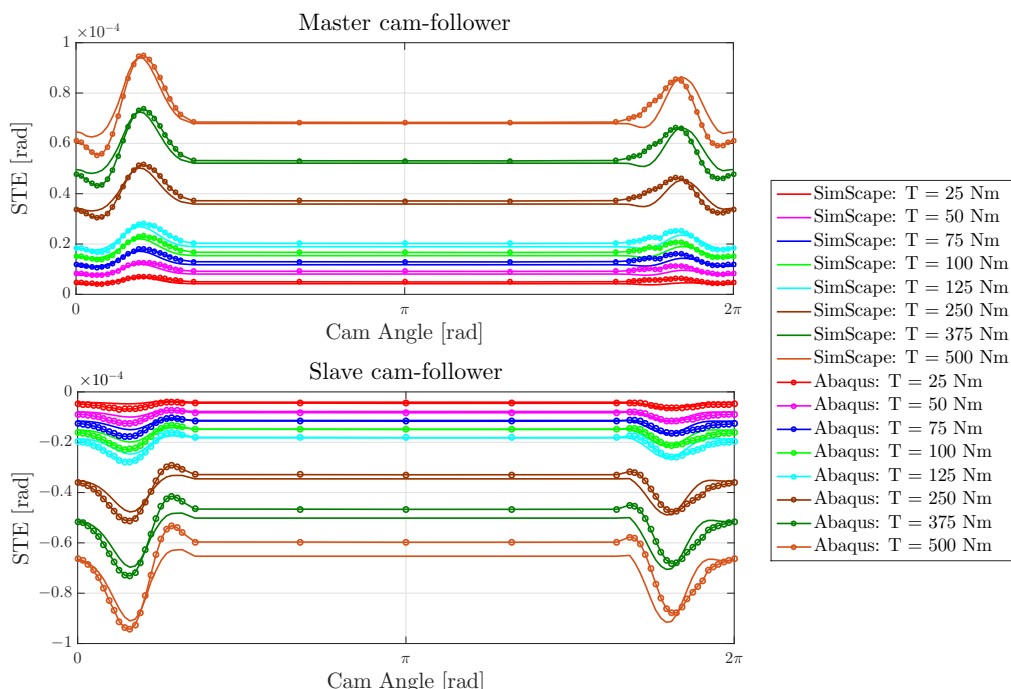

**Figure 6.** STE comparison of the Simscape LPM and FEM for both master (**top**) and slave (**bottom**) cam-follower subsystems at different torque levels.

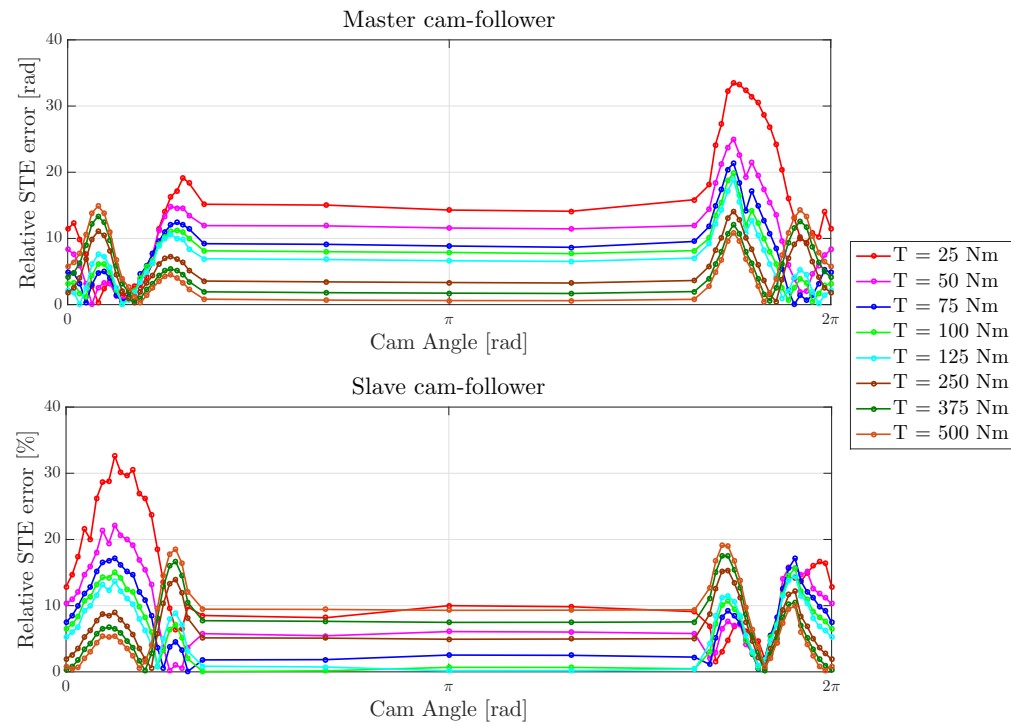

**Figure 7.** Absolute value of the relative STE error comparison of the Simscape LPM with respect to the FEM for both master (**top**) and slave (**bottom**) cam-follower subsystems at different torque levels.

### 4.3. Application of the Optimal Control and Design Optimization Problems

With the underlying model equations are extracted and verified, a classical optimal control problem is next derived and solved with fixed system parameters $p$. Second, multiple co-design study cases are considered in which several model parameters $p$ are optimized.

For both scenarios, the optimization of a periodic profile is considered, i.e., the initial and final condition of the optimized trajectory should be equal. The problem is multi-objective; the minimization of energy consumption (losses) is a trade-off with the minimization of the torques on the follower inertia (a weighted combination). This trade-off is displayed as a Pareto front, with the chosen objective reflecting industrial concerns such as minimization of dynamic loads (leading to fatigue and failure) on the one hand and the requirement of minimizing the system's energy consumption on the other.

### 4.3.1. Optimal Control

The considered cost function for the optimal control problem $\mathcal{J}_{OCP}$ is formulated as follows:

$$\mathcal{J}_{OCP} = \alpha \, c_1 \underbrace{\sum_{i=1}^{N} \zeta_{follower}^2(k)}_{\text{Follower acceleration}} +$$

$$+ (1 - \alpha) \, c_2 T_s \underbrace{\sum_{i=1}^{N} (\omega_{motor}(k) T_{motor}(k))}_{\text{Energy consumption}} + \tag{27}$$

$$+ c_3 \underbrace{\sum_{i=1}^{N} T_{motor}^2(k) + c_4 \sum_{i=1}^{N} \Delta T_{motor}^2(k)}_{\text{Input regularization}} .$$

In the above, $\omega$ denotes the angular velocity and $\zeta$ denotes the angular acceleration. Furthermore, $c_i > 0$ with $i \in [1, 4]$ denote fixed scalar weights which aim to normalize the individual terms of the cost function. These weights are computed by averaging the absolute values of the relative quantities (e.g., $\zeta_{follower}^2$) of the reference forward simulation mentioned above. $\Delta$ denotes the discrete-derivative operator, and $\alpha$ denotes a parameter that moves along the grid $\begin{bmatrix} 0 & 1 \end{bmatrix}$ in six steps, trading off both key objectives, that is, the follower acceleration and the system's energy consumption. The constraints are defined as follows:

$$\text{Equation (23b)} - \text{(23d)};\qquad\qquad\qquad\text{(Model dynamics)} \quad \text{(28a)}$$

$$\theta_{motor}(1) = 0 \text{ [rad]}; \; \theta_{motor}(N) = 2\pi \text{ [rad]};\qquad\text{(Motor constraints)} \quad \text{(28b)}$$

$$\boldsymbol{x}_1 = \boldsymbol{x}_N; \quad \boldsymbol{z}_1 = \boldsymbol{z}_N; \quad \boldsymbol{u}_1 = \boldsymbol{u}_N;\qquad\text{(Periodicity contraints)} \quad \text{(28c)}$$

$$\theta_{follower}(\bar{k}_h) \leq \bar{h}; \quad \theta_{follower}(\underline{k}_h) \geq \underline{h};\qquad\text{(Follower constraints)} \quad \text{(28d)}$$

$$\delta_M(k) \geq 0 \text{ [m]}; \quad \delta_s(k) \geq 0 \text{ [m]}, \quad \forall \, k \in [1, N];\qquad\text{(Penetration constraints)} \quad \text{(28e)}$$

$$\theta_{follower,M}(k) = \theta_{follower,S}(k), \quad \forall \, k \in [1, N];\qquad\text{(Synchronization contraint)} \quad \text{(28f)}$$

$$-250 \text{ [Nm]} \leq T_{motor}(k) \leq 250 \text{ [Nm]}, \quad \forall \, k \in [1, N];\qquad\text{(Peak torque constraints)} \quad \text{(28g)}$$

$$\sqrt{\sum_{i=1}^{N} T_{motor}^2(k)} < 173 \text{ [Nm]}.\qquad\text{(RMS torque constraints)} \quad \text{(28h)}$$

Equation (23b)–(23d) embed the model dynamics of Equation (17a)–(17c) in the optimization problem using a backward Euler integrator. Equation (28b) ensures that one rotation of the system is made. Equation (28c) ensures that the obtained solution is periodic in $\boldsymbol{x}$, $\boldsymbol{z}$, and $\boldsymbol{u}$. Equation (28d) is visually demonstrated in Figure 8. In the figure, it is shown that $\theta_{follower}$ has to be larger than $\bar{h}$ at time sample $\bar{k}_h$ (upper bounds) and, vice versa, $\theta_{follower}$ has to be smaller than $\underline{h}$ at time sample $\underline{k}_h$ (lower bounds). In the figure, a candidate feasible profile for $\theta_{follower}$ is shown. Typically, these lower and upper bounds follow from process constraints of the system to be optimized. Industrial examples include, e.g., weaving (opening and closing of frames during which a yarn is inserted), punching (inserting material, punching the material, removing the material, repeat), etc.

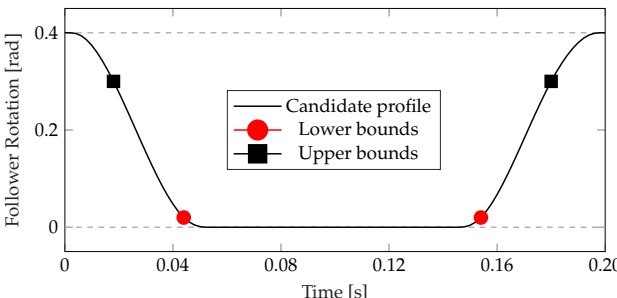

**Figure 8.** The four point constraints on the follower rotation ($\theta_{follower}$). Additionally, a candidate feasible trajectory is shown.

Furthermore, Equation (28e) and (28f) implement the penetration and cam synchronization constraints. Lastly, the motor torque limits are implemented using Equation (28g) and (28h). For the optimal control problem, the sampling frequency is set to 500 Hz (with sampling time $T_s = \frac{1}{500}$ s), and the number of samples is set to $N = 101$. The mean desired motor speed is $10\pi$ rad/s, which is enforced by the constraints in Equation (28b). The initial parameter set is defined in Table 2.

The resulting sparsity patterns for the cost Hessian $\boldsymbol{H} = \frac{d^2 \mathcal{J}_{OCP}}{d^2 \boldsymbol{w}_{opt}}$ and constraint Jacobian $\boldsymbol{J} = \frac{d\boldsymbol{g}}{d\boldsymbol{w}_{opt}}$ are shown in Figure 9 (zoom-plot) with optimization variables $\boldsymbol{w}_{opt}$. It can be seen that the matrices are sparse and block-diagonal due to the multiple shooting transcription method that is applied, allowing their inverses to be calculated efficiently during solving.

The above optimal control problem is executed for $\alpha = \begin{bmatrix} 0, & 0.2, & 0.4, & 0.6, & 0.8, & 1 \end{bmatrix}$. The resulting Pareto front is shown in Figure 10. A clear trade-off between minimization of the follower torques versus minimization of the energy consumption is visible, and a clear "knee-point" (i.e., a sharp angle) is visible. In Figure 11, the time-domain results are shown for $\alpha = 0$ and $\alpha = 1$, respectively. The following observations can be made:

- Periodic results are obtained for both cases in terms of input and state responses.
- The results for the case where $\alpha = 1$ are fairly symmetric over the time axis in terms of motor velocity and follower accelerations, whereas the result for $\alpha = 0$ has more oscillations and shows quite a large peak in follower accelerations at the end of the time horizon, which minimizes energy at the cost of higher accelerations.
- For both cases, the penetration is well above 0. This is caused by the fact that the value of the preload is chosen rather conservatively in order to disallow separation of the contacting bodies, thereby circumventing discontinuities in motion and energy transfer.

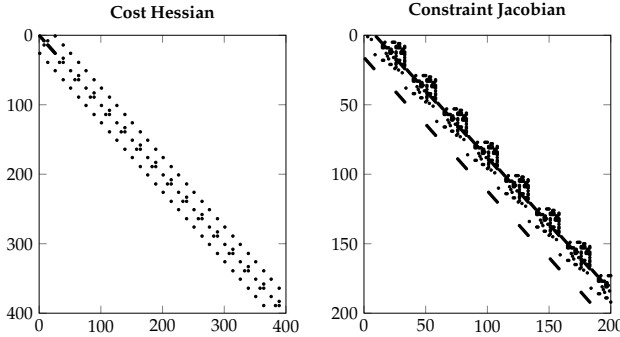

**Figure 9.** Zoom-plot of the block-diagonal cost Hessian and constraint Jacobian.

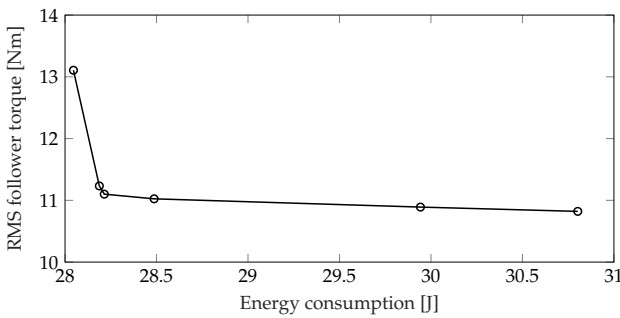

**Figure 10.** Pareto front for the multi-objective OCP.

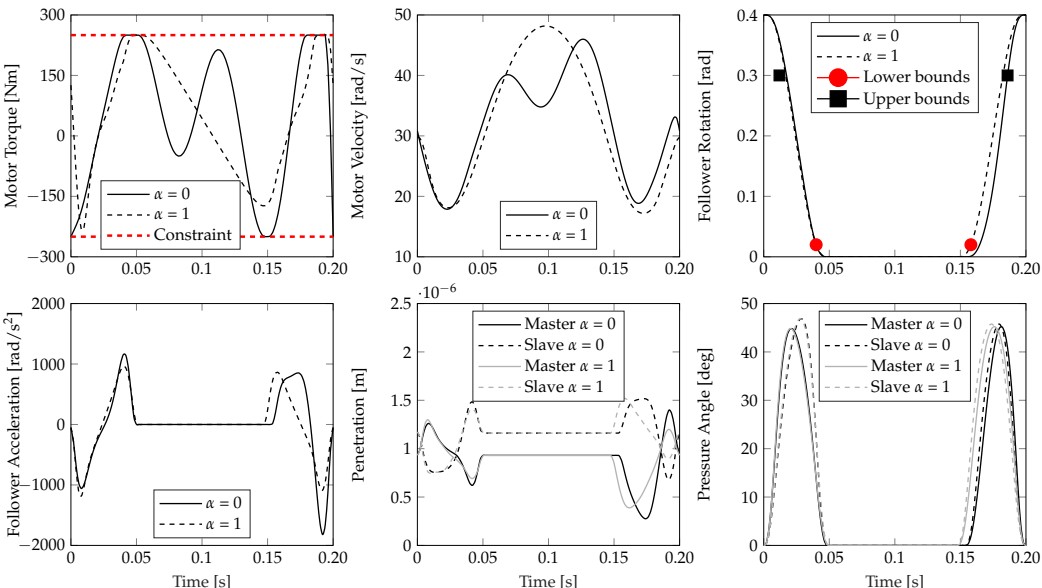

**Figure 11.** OCP results: minimization of energy consumption ($\alpha = 0$) and cam-follower accelerations ($\alpha = 1$).

### 4.3.2. Co-Design

In this section, co-design of the conjugate cam-follower system is considered. In this case, several model parameters $p$ are optimized, along with optimization of the states and control input. We consider four cases of co-design. In each case, additional parameters are optimized. Table 5 sketches the parameter spaces that are explored during the different co-design cases as well as the lower and upper bounds of each parameter and in Appendix B the resulting optimization parameters are reported.

**Table 5.** Tunable parameters along with their lower and upper bounds, noting which parameters are optimized in which co-design cases.

|  | $\alpha_M$ [deg] | $\alpha_S$ [deg] | $\alpha_p$ [rad] | $D_M = D_S$ [m] | $L_M$ [m] | $L_S$ [m] | $\sum \Delta \bar{\theta}_s$ [rad] | $h_1$ [m] | $h_2$ [m] |
|---|---|---|---|---|---|---|---|---|---|
| Lower bound | 110 | 110 | $1 \times 10^{-6}$ | 0.14 | 0.08 | 0.08 | 0 | 0.3 | $-0.1$ |
| Upper bound | 160 | 160 | $1 \times 10^{-4}$ | 0.20 | 0.20 | 0.20 | $2\pi$ | 0.5 | 0.1 |
| Case 1 | × | × | × | × | × | × |  |  |  |
| Case 2 | × | × | × | × | × | × | × |  |  |
| Case 3 | × | × | × | × | × | × | × | × |  |
| Case 4 | × | × | × | × | × | × | × | × | × |

The optimization problem has the same multi-objective cost function as in Equation (27), where the $\alpha$ parameter trades off the energy minimization and cam-follower torque minimization costs. We use the constraints defined in Equation (28a)–(28h) augmented with the following geometrical constraints:

$$\eta_M(k) \leq 35 \text{ [deg]}, \quad \forall\, k \in [1, N];$$
$$\eta_S(k) \leq 35 \text{ [deg]}, \quad \forall\, k \in [1, N];$$
<div align="right">(Pressure angle constraint)   (29a)</div>

$$R_{cam,M}(k) \geq R_{roller,M}, \quad \forall\, k \in [1, N];$$
$$R_{cam,\,S}(k) \leq R_{roller,S}, \quad \forall\, k \in [1, N];$$
<div align="right">(Curvature constraint)   (29b)</div>

$$\sum_{i=1}^{n} \Delta \bar{\theta}_{s,i} = 2\pi.$$
<div align="right">(Sectors constraint)   (29c)</div>

The resulting Pareto fronts are shown in Figure 12, solving each of the four co-design problems for $\alpha = [\,0, 0.2, 0.4, 0.6, 0.8, 1\,]$. In the figure, a clear trade-off between the two objectives is again visible for each of the four cases. Additionally, it can be seen that when more parameters are optimized (as in the later cases), the Pareto front moves towards the origin, i.e., the obtained solutions are more optimal.

Note that the RMS follower torques are somewhat higher for certain $\alpha$ values in co-design case 1 compared to the OCP results shown in Figure 10. This is caused by the additional constraints (see Equation (29a)) introduced for the co-design case; these are not applied in the OCP case, where the design variables are fixed.

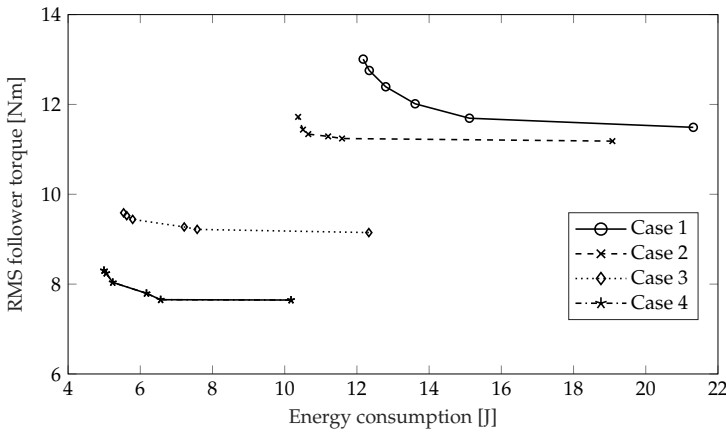

**Figure 12.** Pareto fronts for each of the four multi-objective co-design cases.

Next, the time-domain results for $\alpha = 0$ and $\alpha = 1$ are shown for each of the four cases. Note that in order to preserve space and maintain legibility of the figures, the results for intermediate $\alpha$ are omitted. The following observations can be made:

- Figure 13 shows the follower rotation and follower acceleration. Particularly in case 4, where $h_2$ is additionally optimized, a significant reduction in the follower acceleration (RMS, peak-to-peak) is obtained. For this case, $\theta_{follower}$ "touches" the four-point constraints (Equation (28d)), whereas for in other cases this is not always true.
- Figure 14 shows the resulting motor velocities and motor torques. For $\alpha = 0$, the motor velocity seems to behave in an almost anti-phase manner in cases 3 and 4 compared to the results of the OCP (cases 1 and 2). For $\alpha = 1$, the motor velocity profiles are somewhat flattened in case 3 and 4. For $\alpha = 0$, more oscillations are present in the signals compared to the case where $\alpha = 1$.
- The resulting penetration is shown in Figure 15. It can be seen that the penetration is well above zero for the OCP case, whereas for the co-design cases the penetration profiles move towards the lower bound of 0. This yields a lower energy and required torque for the target follower motion while maintaining contact between the bodies,

thereby circumventing discontinuities in motion and energy transfer due to the applied lower bound.

- The resulting shapes of the master and follower cams for each of the four co-design cases are shown in Figure 16 along with the reference design used for the OCP. In general, smoother geometries are obtained for higher case numbers compared to the reference design, which is due to the applied constraints on follower displacement (Equation (28d)). For co-design cases 1 and 2, the shapes remain rather similar to the reference design (OCP), although the absolute sizes are increased for both the master and the slave cam. For cases 3 and 4, the solutions lead to different cam designs overall due to the additional freedom in $p$.

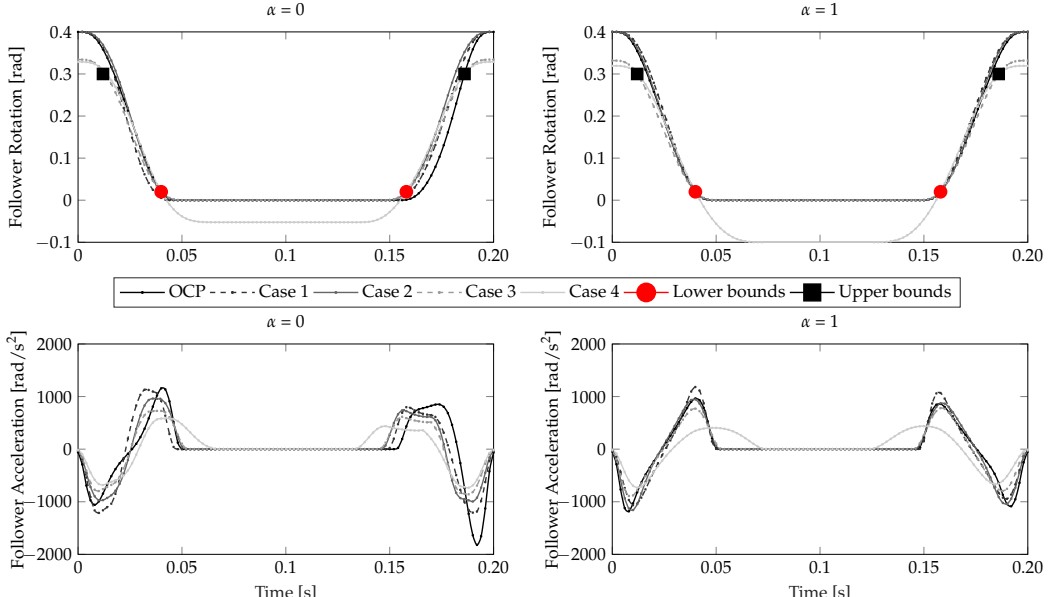

**Figure 13.** Co-design result: follower rotation and accelerations.

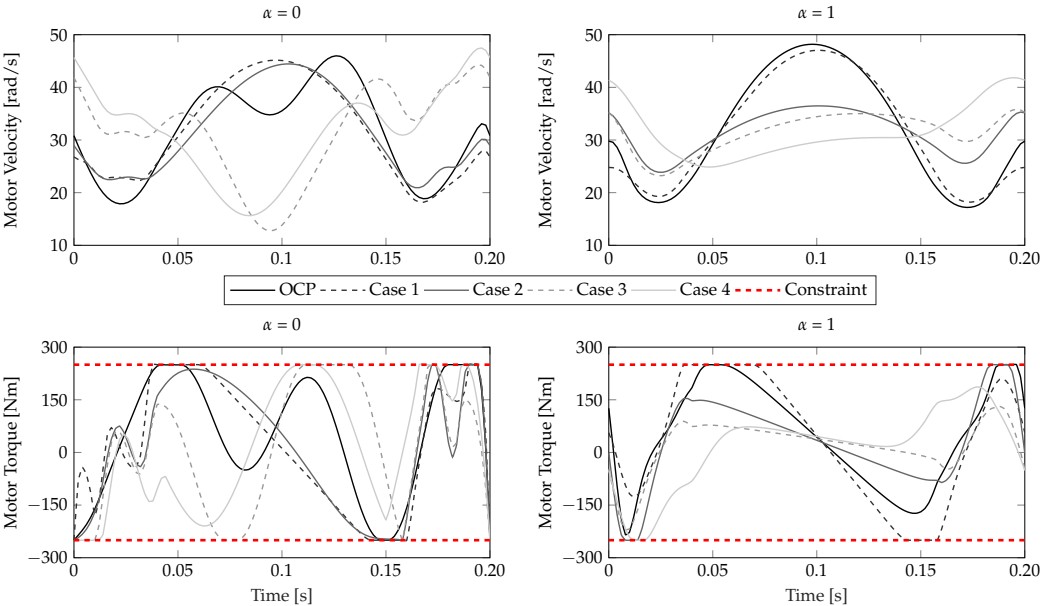

**Figure 14.** Co-design result: motor speed and torque.

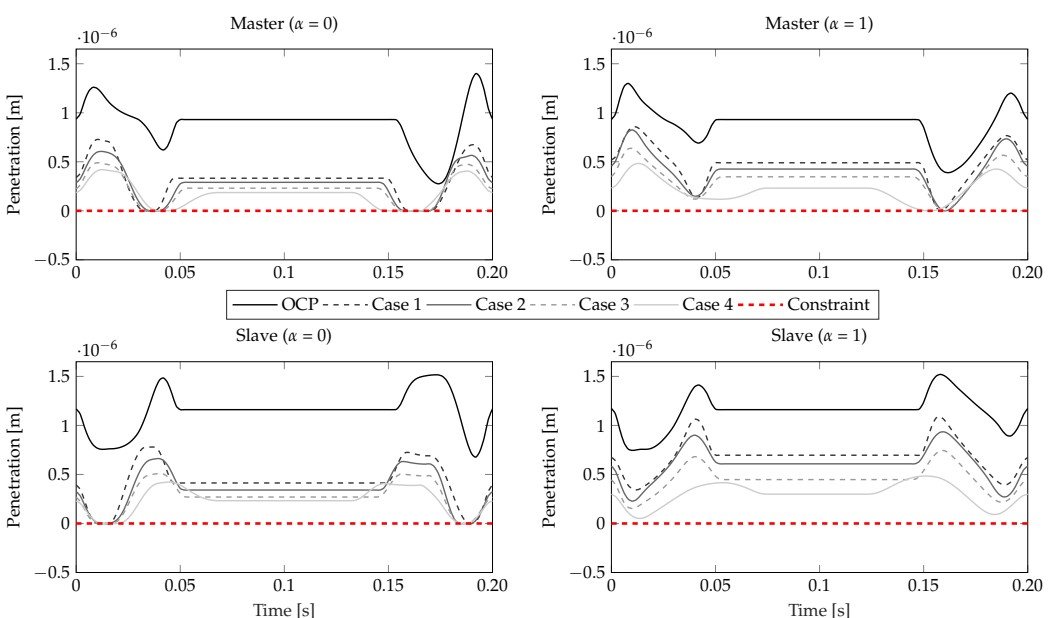

**Figure 15.** Co-design result: roller-cam bodies penetration.

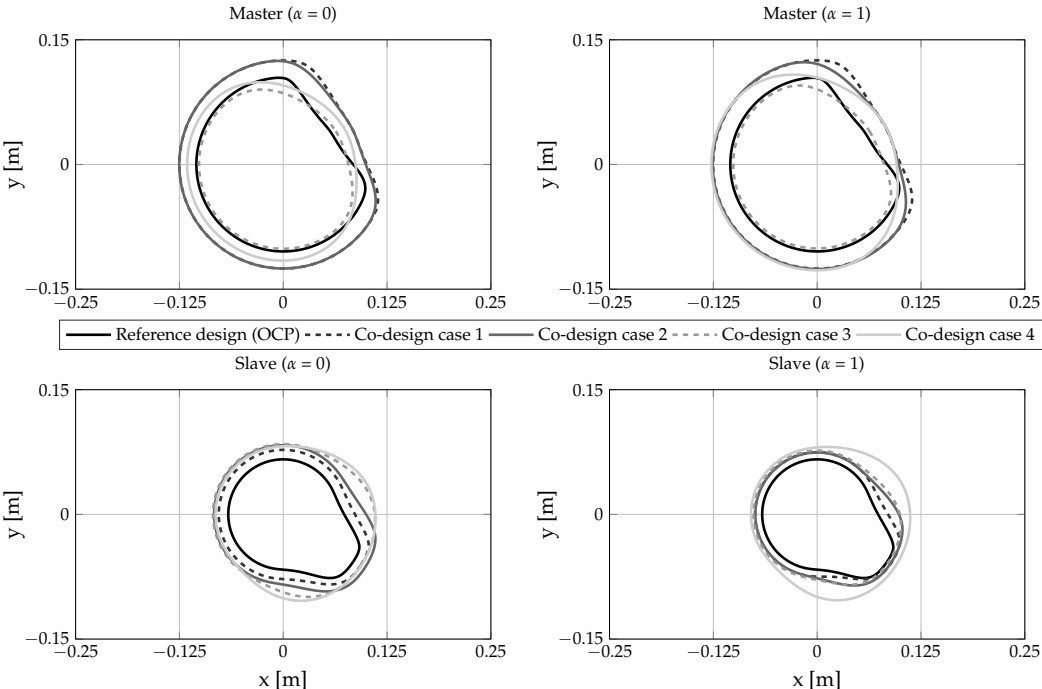

**Figure 16.** The resulting cam shapes for the master and follower cams for the reference design (OCP) and the four co-design cases for $\alpha = \begin{bmatrix} 0 & 1 \end{bmatrix}$.

## 5. Conclusions and Future Work

In this work, an integrated model-based co-design optimization toolchain called the DriveTrain Co-Design Toolchain is presented. The proposed toolchain reduces the required modelling effort by interfacing with MATLAB Simscape and the need for expert knowledge by defining and solving optimization problems. In this paper, the proposed toolchain is successfully applied to a mechatronic drivetrain system involving a high-fidelity and validated 1D conjugate cam-follower model. The toolchain is generally applicable to various other types of drivetrains or mechatronic systems as well.

In particular, when dealing with model-based co-design problems, high-fidelity parameterized models are of paramount importance in obtaining reliable information/data as

result of optimization processes which can be exploited in the different phases of product design. Similarly, the proper selection of the cost function and the constraints for the considered use case is key to obtaining useful results. Both of these elements have been successfully embraced in this work to show the robustness of the proposed Co-design Toolchain for industrially relevant applications.

Our future work will focus on extending the proposed methodology to systems with greater complexity (e.g., more states or more complex loads). Additionally, the optimization toolchain will be coupled with other modelling interfaces, such as Modelica [45] or Simscape Multibody [46], allowing for broader industrial uptake. Moreover, the tool has the potential to be augmented with several additional functionalities: (i) multi-stage optimization, allowing the design to be optimized for multiple operating conditions in a single optimization problem; (ii) iterative learning control, allowing optimal control of a system in cases where only approximate models are available; and (iii) model-predictive control, a step towards implementation/deployment of the OCP on a physical system).

**Author Contributions:** Conceptualization and implementation, R.A. and J.W.; methodology, R.A., J.W., E.K. and J.G.; numerical validation, R.A.; resources, W.D.; writing, review, and editing, R.A., J.W., E.K., J.G. and J.C.; supervision J.C. and W.D. All authors have read and agreed to the published version of the manuscript.

**Funding:** This work has been carried out within the framework of the projects Flanders Make ICON: Physical and control co-design of electromechanical drivetrains for machines and vehicles (DriveTrain Co-Design), Flanders Make ICON: Selection, design and control of electromagnetic torque ripple reduction for drivetrains (Torque-Ripple Reduction), and KU Leuven-BOF PFV/10/002 Centre of Excellence: Optimization in Engineering (OPTEC). The research was partially supported by Flanders Make: the Flemish strategic research centre for the manufacturing industry.

**Institutional Review Board Statement:** Not applicable.

**Informed Consent Statement:** Not applicable.

**Data Availability Statement:** The data presented within this study are resulting from activities within the acknowledged projects and are available therein.

**Conflicts of Interest:** The Authors declare no conflict of interest.

## Abbreviations

The following abbreviations are used in this manuscript:

| | |
|---|---|
| FEM | Finite Element Method |
| LPM | Lumped Paramter Model |
| DC | Direct Current |
| n-D | n-dimensional space |
| DAE | Differential Algebraic Equation |
| ODE | Ordinary Differential Equation |
| OCP | Optimal Control Problem |
| NLP | Non-Linear Program |
| MEMS | Micro-Electronic Mechanical System |
| MPC | Model Predictive Control |
| PID | Proportional Integral Derivative |
| CAD | Computer-Aided Design |
| STE | Static Transmission Error |
| MPC | Multi-Point Contraint |
| $\mathbb{Z}$ | integer numbers set |
| $\mathbb{R}$ | real numbers set |
| $a, A \in \mathbb{R}$ | scalar |
| $\boldsymbol{a} \in \mathbb{R}^{n_a}$ | column vector |
| $\overrightarrow{\square} \in \mathbb{R}^3$ | 3D vector |

| $\hat{\square} \in \mathbb{R}^3$ | 3D unit vector |
|---|---|
| $\{\square\} \in \mathbb{R}^{3\times3}$ | right-handed orthonormal axes system |
| $\boldsymbol{A} \in \mathbb{R}^{n_1 \times n_2}$ | matrix |
| $\underline{\square}$ | lower bound |
| $\bar{\square}$ | upper bound |
| $\square^{-1}$ | inverse matrix operator |
| $\square_k = \square(t = t_k)$ | $k^{th}$ time step |
| $\dot{\square} = \frac{d\square}{dt}, \ddot{\square} = \frac{d^2\square}{dt^2}$ | time derivatives |
| $\frac{da_1}{da_2} \in \mathbb{R}^{n_{a_1} \times n_{a_2}}$ | total derivative |
| $\frac{\partial a_1}{\partial a_2} \in \mathbb{R}^{n_{a_1} \times n_{a_2}}$ | partial derivative |

## Appendix A. Analytical Inverse Kinematic Solution of the Cam-Follower System

In this appendix, the analytical solution of the inverse kinematics is presented and discussed for planar conjugate cam-follower systems, generalizing the approach proposed in [47].

In particular, the overall motion of the cam-follower system is solved assuming the cam to be fixed while the follower center rotates rigidly around the cam center. As shown in Figure 3, for a given distance between the cam and follower axis $OA$, a certain lever arm length $AP$, roller radius $R_{roller}$, and cam-follower contact point $OC(\theta)$ can be determined.

Figure 3a indicates the main geometrical features, while in Figure 3b the conjugate cam-follower system for a generic $\theta$ value is shown. To better describe the problem the variables $\psi$ and $\psi'$ are introduced:

$$\psi = -\theta + f(\theta) + \alpha_o. \tag{A1}$$

Moreover, the time derivative of Equation (A1) is computed by applying the chain rule:

$$\dot{\psi} = -\dot{\theta} + f_d(\theta)\dot{\theta}. \tag{A2}$$

The variables $\psi$ and $\dot{\psi}$ describe the desired follower kinematics; $\alpha_o$ is an additional follower orientation parameter, as shown in Figure 3a.

Similarly,

$$\psi' = -\theta + \gamma + \alpha_o + \alpha_p; \tag{A3a}$$

$$\dot{\psi}' = -\dot{\theta} + \dot{\gamma}. \tag{A3b}$$

Here, $\psi'$ and $\dot{\psi}'$ define the "dynamic" follower motion equations. The parameter $\alpha_p$ represents the angular pre-load that generally is introduced into the system to ensure continuous contact between the roller and the cam. After the variables $\psi$ and $\psi'$ and the time derivatives are established, the inverse kinematics of the cam-follower system can be computed.

In the following section, the different steps to compute the inverse kinematics are reported. In the notation, no distinction is made between the master and slave cam-follower subsystems, as the same equation holds for both.

### Appendix A.1. Step 1: Calculation of the Pitch Point P and Its Derivatives Expressed in the Global Frame

Based on Figure 3, the position and velocity of the pitch point can be computed as follows:

$$OP_x = D\cos(\theta) + L\cos(\psi); \tag{A4a}$$

$$OP_y = -D\sin(\theta) + L\sin(\psi); \tag{A4b}$$

$$\dot{OP}_x = -D\sin(\theta)\dot{\theta} - L\sin(\psi)\dot{\psi}; \tag{A4c}$$

$$\dot{OP}_y = -D\cos(\theta)\dot{\theta} + L\cos(\psi)\dot{\psi}. \tag{A4d}$$

*Appendix A.2. Step 2: Compute the Angle $\phi_n$ and Its Derivative*

The angle $\phi_n$ represents the angle between the global $x$ axis and the normal $\hat{n}$ axis. The orientation of the local $\hat{t} - \hat{n}$ frame is required in order to derive the location contact point and its derivative. It can be computed by means of the orientation of the velocity vector $\dot{OP}$:

$$\phi_t = \tan^{-1}\left(\frac{\dot{OP}_y}{\dot{OP}_x}\right); \tag{A5a}$$

$$\phi_n = \phi_t + \frac{\pi}{2}. \tag{A5b}$$

The time derivatives of Equation (A5a) and (A5b) become

$$\dot{\phi}_t = \dot{\phi}_n = \frac{\ddot{OP}_y\dot{OP}_x - \ddot{OP}_x\dot{OP}_y}{\dot{OP}_x^2\left[1 + \left(\frac{\dot{OP}_y}{\dot{OP}_x}\right)^2\right]}. \tag{A6}$$

Equation (A6) requires the second-order derivatives of the pitch point that would lead to higher-order differential equations to be solved through an augmented system states. In this regard, we propose an efficient solution that preserves both system accuracy and size.

By applying the chain rule, $\dot{\phi}_n$ can be written as follows:

$$\dot{\phi}_t = \frac{\partial \phi_t}{\partial \theta}\frac{d\theta}{dt} = \frac{\partial \phi_t}{\partial \theta}\dot{\theta}; \tag{A7a}$$

$$\dot{\phi}_n = \frac{\partial \phi_n}{\partial \theta}\frac{d\theta}{dt} = \frac{\partial \phi_n}{\partial \theta}\dot{\theta}. \tag{A7b}$$

Thus, with the left-hand side of Equation (A7b), it holds that

$$\frac{\partial \phi_t}{\partial \theta} = \frac{\partial \phi_n}{\partial \theta} = \frac{\partial \Phi}{\partial \theta}. \tag{A8}$$

Assuming that the cam and follower bodies are considered to be rigid while allowing a small local compliance $\delta$ in the contact area, the variation of the angles $\phi_t$ and $\phi_n$ with respect to the cam angle $\theta$ can be considered independently of the system dynamics. Therefore, the kinematic variable $\Phi$ is introduced and expressed as follows:

$$\Phi = \phi_t|_{\dot{\theta}=1} = \tan^{-1}\left(\frac{\Lambda}{\Gamma}\right), \tag{A9}$$

with

$$\Lambda = \dot{OP}_y|_{\dot{\theta}=1} = -D\cos(\theta) + L\cos(\psi)(-1 + f_d); \tag{A10a}$$

$$\Gamma = \dot{OP}_x|_{\dot{\theta}=1} = -D\sin(\theta) - L\sin(\psi)(-1 + f_d); \tag{A10b}$$

$$\frac{\partial \Lambda}{\partial \theta} = D\sin(\theta) - L\sin(\psi)(-1 + f_d)^2 + f_{dd}L\cos(\psi); \tag{A10c}$$

$$\frac{\partial \Gamma}{\partial \theta} = -D\cos(\theta) - L\cos(\psi)(-1 + f_d)^2 - f_{dd}L\sin(\psi). \tag{A10d}$$

Similarly to Equation (A6), Equation (A8) can be expressed as

$$\frac{\partial \Phi}{\partial \theta} = \frac{\frac{\partial \Lambda}{\partial \theta}\Gamma - \frac{\partial \Gamma}{\partial \theta}\Lambda}{\Gamma^2 \left[1 + \left(\frac{\Lambda}{\Gamma}\right)^2\right]}. \tag{A11}$$

The curvature radius of the cam profile $R_{cam}$ is a purely geometrical parameter, and as such is independent of the system dynamics. Therefore, it can be expressed as function of the cam angle $\theta$ as

$$R_{cam} = \frac{(\Lambda^2 + \Gamma^2)^{3/2}}{|\frac{\partial \Lambda}{\partial \theta}\Gamma - \frac{\partial \Gamma}{\partial \theta}\Lambda|}. \tag{A12}$$

*Appendix A.3. Step 3: Compute the Kinematic Contact Point C and Its Derivative with Respect to the Follower Center A*

In the previous steps, the coordinates of the pitch point $P$ and the orientation of the local $\hat{t} - \hat{n}$ frame were derived. From these quantities, the kinematic contact point $C$ with respect to the follower center $A$ can be determined using the following geometrical projection:

$$AC_x = L\cos(\psi) + R_{roller}\cos(\phi_n); \tag{A13a}$$
$$AC_y = L\sin(\psi) + R_{roller}\sin(\phi_n); \tag{A13b}$$
$$\dot{AC}_x = -L\sin(\psi)\dot{\psi} - R_{roller}\sin(\phi_n)\dot{\phi}_n; \tag{A13c}$$
$$\dot{AC}_y = L\cos(\psi)\dot{\psi} + R_{roller}\cos(\phi_n)\dot{\phi}_n. \tag{A13d}$$

*Appendix A.4. Step 4: Compute the Non-Kinematic Contact Point C′ and Its Derivative with Respect to the Follower Center A*

As we have allowed a local compliance $\delta$, which is treated as an additional degree of system freedom, we have not yet satisfied the desired kinematics. Therefore, the follower motion is driven by the contact forces and the local compliance expressed as the difference between the perfect kinematic motion and the non-kinematic motion. In this regard, the cam point $C'$ represents the roller contact point in the presence of a small penetration $\delta$ into the cam body. Similarly to step 3 in Appendix A.3, the calculation of $AC$ and $\dot{AC}$ is performed considering $\dot{\psi}'$ and $\psi'$ instead of $\dot{\psi}'$ and $\psi'$. It is assumed that the orientation of the local $\hat{t} - \hat{n}$ frame as compared to the pure kinematic motion does not change. This assumption is valid if the joint positions of the cam, follower, and roller can be considered fixed.

*Appendix A.5. Step 5: Transform All Variables to the Local Frame*

Thus far, the computed quantities have been expressed with respect to the global $x$-$y$ frame, while the contact stiffness (or compliance) and damping relationships have been described according to the normal and tangent axis ($\hat{t} - \hat{n}$ frame). Here, $AC$ and $\dot{AC}$ are projected onto the $\hat{t} - \hat{n}$ frame, as follows:

$$\begin{bmatrix} AC_t \\ AC_n \end{bmatrix} = \begin{bmatrix} \cos(\phi_t) & \sin(\phi_t) \\ -\sin(\phi_t) & \cos(\phi_t) \end{bmatrix} \begin{bmatrix} AC_x \\ AC_y \end{bmatrix}; \tag{A14a}$$

$$\begin{bmatrix} \dot{AC}_t \\ \dot{AC}_n \end{bmatrix} = \begin{bmatrix} \cos(\phi_t) & \sin(\phi_t) \\ -\sin(\phi_t) & \cos(\phi_t) \end{bmatrix} \begin{bmatrix} \dot{AC}_x \\ \dot{AC}_y \end{bmatrix} + \begin{bmatrix} -\sin(\phi_t) & \cos(\phi_t) \\ -\cos(\phi_t) & -\sin(\phi_t) \end{bmatrix} \begin{bmatrix} \dot{AC}_x \\ \dot{AC}_y \end{bmatrix}\dot{\phi}_t. \tag{A14b}$$

*Appendix A.6. Step 6: Compute the Contact Point Velocity and Project it onto the Tangential Axis*

The cam velocity at the contact point along the tangential axis $\hat{t}$ is required in order to calculate the roller slip velocity in Equation (16). First, the pitch point velocity in the global axis system is calculated using Equation (A4c) and (A4d):

$$OC_x = OP_x + R_{roller} \cos(\phi_n); \tag{A15a}$$

$$OC_y = OP_y + R_{roller} \sin(\phi_n); \tag{A15b}$$

$$\dot{OC}_x = \dot{OP}_x - R_{roller} \sin(\phi_n)\dot{\phi}_n; \tag{A15c}$$

$$\dot{OC}_y = \dot{OP}_y + R_{roller} \cos(\phi_n)\dot{\phi}_n. \tag{A15d}$$

when the contact point position and velocity are known, they can be projected onto the tangential axis $\hat{t}$, obtaining

$$OC_t = -OC_x \sin(\phi_n) + OC_y \cos(\phi_n); \tag{A16a}$$

$$\dot{OC}_t = -\dot{OC}_x \sin(\phi_n) - OC_x \cos(\phi_n)\dot{\phi}_n + \dot{OC}_y \cos(\phi_n) - OC_y \sin(\phi_n)\dot{\phi}_n. \tag{A16b}$$

Finally the angle $\eta$ between the direction of motion of the follower and the direction of the axis of transmission is computed. This angle is known as the pressure angle $\eta$, and is a design parameter that needs to be kept as small as possible.

$$\eta = \arccos\left(\frac{\cos(\phi_n)OC_x + \sin(\phi_n)OC_y}{\sqrt{OC_x^2 + OC_y^2}}\right). \tag{A17}$$

## Appendix B. Design Parameters Resulting from the Optimization Solutions

In Tables A1–A4, the (optimized) parameter values are shown for each of the four considered co-design cases.

**Table A1.** The resulting (optimized) parameter values of co-design case 1 given variable weight $\alpha$.

|  | $\alpha_M$ [deg] | $\alpha_S$ [deg] | $\alpha_p$ [rad] | $D_M = D_S$ [m] | $L_M$ [m] | $L_S$ [m] |
|---|---|---|---|---|---|---|
| LB | 110 | −160 | $10^{-6}$ | 0.14 | 0.08 | 0.08 |
| UB | 160 | −110 | $10^{-4}$ | 0.20 | 0.20 | 0.20 |
| $\alpha = 0$ | 110 | −130.2 | $4.54 \times 10^{-6}$ | 0.171 | 0.08 | 0.0857 |
| $\alpha = 0.2$ | 110 | −130.09 | $4.56 \times 10^{-6}$ | 0.171 | 0.08 | 0.0860 |
| $\alpha = 0.4$ | 110 | −128.26 | $4.60 \times 10^{-6}$ | 0.171 | 0.08 | 0.0867 |
| $\alpha = 0.6$ | 110 | −130.02 | $4.85 \times 10^{-6}$ | 0.171 | 0.08 | 0.0859 |
| $\alpha = 0.8$ | 110 | −130.02 | $5.17 \times 10^{-6}$ | 0.171 | 0.08 | 0.0859 |
| $\alpha = 1$ | 110 | −132.75 | $7.6 \times 10^{-6}$ | 0.171 | 0.08 | 0.08 |

**Table A2.** The resulting (optimized) parameter values of co-design case 2 given variable weight $\alpha$.

| | $\alpha_M$ [deg] | $\alpha_S$ [deg] | $\alpha_p$ [rad] | D [m] | $L_M$ [m] | $L_S$ [m] | $\Delta\bar{\theta}_s$ [rad] |
|---|---|---|---|---|---|---|---|
| LB | 110 | −160 | $10^{-6}$ | 0.14 | 0.08 | 0.08 | 0 |
| UB | 160 | −110 | $10^{-4}$ | 0.20 | 0.20 | 0.20 | $2\pi$ |
| $\alpha = 0$ | 110 | −126.45 | $3.79 \times 10^{-6}$ | 0.171 | 0.08 | 0.0871 | [1.23, 3.71, 1.33] |
| $\alpha = 0.2$ | 110 | −124.79 | $3.95 \times 10^{-6}$ | 0.172 | 0.08 | 0.0819 | [1.25, 3.59, 1.42] |
| $\alpha = 0.4$ | 110 | −126.18 | $4.0 \times 10^{-6}$ | 0.172 | 0.08 | 0.0812 | [1.25, 3.59, 1.43] |
| $\alpha = 0.6$ | 110 | −126.88 | $4.14 \times 10^{-6}$ | 0.172 | 0.08 | 0.0831 | [1.35, 3.38, 1.55] |
| $\alpha = 0.8$ | 110 | −128.61 | $4.2 \times 10^{-6}$ | 0.172 | 0.08 | 0.0867 | [1.36, 3.34, 1.57] |
| $\alpha = 1$ | 110 | −133.01 | $6.64 \times 10^{-6}$ | 0.172 | 0.08 | 0.08 | [1.42, 3.28, 1.57] |

**Table A3.** The resulting (optimized) parameter values of co-design case 3 given variable weight $\alpha$.

| | $\alpha_M$ [deg] | $\alpha_S$ [deg] | $\alpha_p$ [rad] | D [m] | $L_M$ [m] | $L_S$ [m] | $\Delta\bar{\theta}_s$ [rad] | $h_1$ [m] |
|---|---|---|---|---|---|---|---|---|
| LB | 110 | −160 | $10^{-6}$ | 0.14 | 0.08 | 0.08 | 0 | 0.3 |
| UB | 160 | −110 | $10^{-4}$ | 0.20 | 0.20 | 0.20 | $2\pi$ | 0.5 |
| $\alpha = 0$ | 112.08 | −125.58 | $3.08 \times 10^{-6}$ | 0.147 | 0.08 | 0.083 | [1.74, 2.38, 2.15 ] | 0.334 |
| $\alpha = 0.2$ | 112.04 | −126.1 | $3.14 \times 10^{-6}$ | 0.147 | 0.08 | 0.083 | [1.75, 2.30, 2.21 ] | 0.333 |
| $\alpha = 0.4$ | 112.17 | −126.17 | $3.2 \times 10^{-6}$ | 0.147 | 0.08 | 0.083 | [1.76, 2.20, 2.31 ] | 0.332 |
| $\alpha = 0.6$ | 112.31 | −124.41 | $3.15 \times 10^{-6}$ | 0.147 | 0.08 | 0.084 | [1.29, 3.11, 1.87] | 0.330 |
| $\alpha = 0.8$ | 112.37 | −122.81 | $3.2 \times 10^{-6}$ | 0.147 | 0.08 | 0.086 | [1.32, 3.07, 1.88 ] | 0.330 |
| $\alpha = 1$ | 112.25 | −131.32 | $5.03 \times 10^{-6}$ | 0.147 | 0.08 | 0.08 | [1.40, 3.09, 1.77] | 0.331 |

**Table A4.** The resulting (optimized) parameter values of co-design case 4 given variable weight $\alpha$.

| | $\alpha_M$ [deg] | $\alpha_S$ [deg] | $\alpha_p$ [rad] | D [m] | $L_M$ [m] | $L_S$ [m] | $\Delta\bar{\theta}_s$ [rad] | $h_1$ [m] | $h_2$ [m] |
|---|---|---|---|---|---|---|---|---|---|
| LB | 110 | −160 | $10^{-6}$ | 0.14 | 0.08 | 0.08 | 0 | 0.3 | −0.1 |
| UB | 160 | −110 | $10^{-4}$ | 0.20 | 0.20 | 0.20 | $2\pi$ | 0.5 | 0.1 |
| $\alpha = 0$ | 111.15 | −124.8 | $2.63 \times 10^{-6}$ | 0.157 | 0.08 | 0.08 | [2.26, 1.49, 2.52] | 0.381 | −0.052 |
| $\alpha = 0.2$ | 111.08 | −125.25 | $2.67 \times 10^{-6}$ | 0.158 | 0.08 | 0.08 | [2.28, 1.43, 2.56] | 0.383 | −0.054 |
| $\alpha = 0.4$ | 111.07 | −124.21 | $2.64 \times 10^{-6}$ | 0.161 | 0.08 | 0.08 | [2.29, 1.35, 2.63 ] | 0.394 | −0.067 |
| $\alpha = 0.6$ | 110.88 | −122.98 | $2.62 \times 10^{-6}$ | 0.164 | 0.08 | 0.08 | [2.09, 1.62, 2.56] | 0.409 | −0.086 |
| $\alpha = 0.8$ | 110.83 | −121.83 | $2.69 \times 10^{-6}$ | 0.166 | 0.08 | 0.08 | [2.14, 1.4, 2.70 ] | 0.420 | −0.099 |
| $\alpha = 1$ | 110.91 | −123.52 | $3.38 \times 10^{-6}$ | 0.166 | 0.08 | 0.08 | [2.18, 1.45, 2.64] | 0.419 | −0.1 |

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
