# Peer review of "An Integrated Co-Design Optimization Toolchain Applied to a Conjugate Cam-Follower Drivetrain System"

_machines, doi:10.3390/machines11040486_

Round 1

Reviewer 1 Report

The paper presents an optimization approach to controlling a system with uncertain model on the basis of drivetrain system. IN the introductory part of the apper, the authors present a co-design idea, based on examples of simultaneous control and optimzation tasks. A step-by-step procedure is presented to obtaining a solution of the problem on the basis of a Simscape. Novelty is given, contribution as well. 

In the next section, the authors present a cam-followe. Table 1 - why is 7th order given? Any calculation campaign performed to set this degree of a polynomial? Any tradeoff between accuracy and simplicity of calculations? 

Section 3 is very interesting, as it outlines the optimizaton procedure. Formula (25) should provide lower and upper bounds, and there is some problem with a lower bound symbol. 

FOrmula(27) - any methodology behind selection of c_i's? How to make the linear combination elements of equal amplitude? Or comparable amplitude, shoudl c_i's have the same meaning or amplitude? 

Figure 9  - how has Pareto front been identified? is it just a single front or one of many? 

Reviewer 2 Report

    The contribution has a precisely defined structure. Meets the conditions for such contributions This article is highly technical. Graphically, the presentation of pictures and graphs is very good. The description of the pictures is also very good, clear and of high quality.  

The authors of the article present an awful lot of results simulated in the Matlab program.  

I am not an expert on the Matlab program, but if another reviewer confirms that the entered equations and initial conditions in the Matlab program are correct, then the article meets the conditions for publication.  

The used literature is used correctly, and it seems that the authors have an overview in the given field.  

I think that the conclusion is very extensive and it might be better to simplify it a bit. Also, the scope of the article is very large, it is more than 30 pages. Articles of this type should have a range of 20 +-4 pages.  

It's just my advice.  

I recommend that this article be published after a Matlab expert has checked it.  

It can be seen that the authors are experts in the given field and present results intended for a specific group of experts working in the given field.

Reviewer 3 Report

This work presents an optimization toolchain that interfaces with MATLAB Simscape to optimize high-fidelity drivetrain models. The toolchain reduces user effort, skill, and computation time required for optimization. The toolchain is illustrated on an industrially relevant conjugate cam-follower system, which is modeled in the Simscape environment and validated with respect to a higher-fidelity modeling technique FEM. The tool Simscape2CasADi is developed to extract symbolic equations from the Simscape model. The paper is well written and I suggest acceptance after the following two minor revision.

1, Please provide more information about the step type and boundary conditions of the FEM model, as well as the input and output parameters.

2, Please add the error percentage and explain the possible reason for the error when comparing the method with FEM.

Reviewer 4 Report

I appreciate the authors' contribution to the scientific field. Although the manuscript is well written, I do, however, have some suggestions.

- I suggest you provide more context on the challenges and limitations of machine builders and automotive companies achieving higher performance, regulations and competition rather than just general statements and trends of increasing requests for innovative and performant manufacturing products.

- What are the challenges and limitations of conventional prototype-based techniques in designing and optimizing cam-follower systems? How do physics-based simulation models address these challenges?

- Clarify the significance and novelty of the ad-hoc 1D dual cam-follower model developed in Simscape. How does it differ from existing models, and why is it better suited for concurrent optimization?

- Provide a more detailed explanation of the system-level architecture of the cam-follower, including how the input torque, damping elements, and inertia element interact with the conjugate cam-follower element?

Overall, this part of the manuscript could benefit from more detailed explanations and a more explicit focus on the specific contributions and significance of the study.

Also, throughout the text, you reference "Fig. 1" and later on "Figure 4" in the text. I think this should be appropriately referenced. This also holds for Figure 8. Also, Figure 15, I cannot find the referencing throughout the text.
